# Greening of the Sahara suppressed ENSO activity during the mid-Holocene

Francesco S.R. Pausata[1,2], Qiong Zhang[3], Francesco Muschitiello[4,5,6], Zhengyao Lu[7], Léon Chafik[8], Eva M. Niedermeyer[9], J. Curt Stager[10], Kim M. Cobb[11] & Zhengyu Liu[7,12]

The evolution of the El Niño-Southern Oscillation (ENSO) during the Holocene remains uncertain. In particular, a host of new paleoclimate records suggest that ENSO internal variability or other external forcings may have dwarfed the fairly modest ENSO response to precessional insolation changes simulated in climate models. Here, using fully coupled ocean-atmosphere model simulations, we show that accounting for a vegetated and less dusty Sahara during the mid-Holocene relative to preindustrial climate can reduce ENSO variability by 25%, more than twice the decrease obtained using orbital forcing alone. We identify changes in tropical Atlantic mean state and variability caused by the momentous strengthening of the West Africa Monsoon (WAM) as critical factors in amplifying ENSO's response to insolation forcing through changes in the Walker circulation. Our results thus suggest that potential changes in the WAM due to anthropogenic warming may influence ENSO variability in the future as well.

[1] Department of Meteorology, Stockholm University and Bolin Centre for Climate Research, Stockholm 10691, Sweden. [2] Department of Earth and Atmospheric Sciences, University of Quebec in Montreal (UQÀM), Montreal, Quebec, Canada H3C 3P8. [3] Department of Physical Geography, Stockholm University and Bolin Centre for Climate Research, Stockholm 10691, Sweden. [4] Department of Geological Sciences, Stockholm University and Bolin Centre for Climate Research, Stockholm 10691, Sweden. [5] Lamont-Doherty Earth Observatory of Columbia University, Palisades, New York 10964, USA. [6] Uni Research Climate and Bjerknes Centre for Climate Research, Bergen 5007, Norway. [7] LaCOAS, School of Physics, Peking University, Beijing 100871, China. [8] Geophysical Institute and Bjerknes Centre for Climate Research, Bergen 5007, Norway. [9] Senckenberg Biodiversity and Climate Research Centre (BiK-F), Frankfurt am Main 60325, Germany. [10] Natural Sciences Division, Paul Smith's College, Paul Smiths, New York 12970, USA. [11] School of Earth and Atmospheric Sciences, Georgia Institute of Technology, Atlanta, Georgia 30332, USA. [12] Department of Atmospheric and Oceanic Sciences & Center for Climatic Research, University of Wisconsin-Madison, Madison, Wisconsin 53706, USA. Correspondence and requests for materials should be addressed to F.S.R.P. (email: francesco.pausata@misu.su.se).

The El Niño-Southern Oscillation (ENSO) system is a major component of tropical climate variability and is known to affect climate worldwide[1]. Both observations and climate model simulations suggest that changes in ENSO behaviour may eventually occur under the current global warming trend[2–6]. Therefore, it is crucial to understand the nature and causes of past ENSO variability to constrain potential future changes. Paleoclimate archives such as marine[7] and lake[8] sediment records, foraminifera[9] from eastern equatorial Pacific, laminated lake deposits in Ecuador[10,11], and fossil corals[12,13] from western equatorial Pacific suggest that ENSO variability may have been reduced by 30–60% (refs 12,14,15) during the mid-Holocene (MH, ~4–7,000 years BP) relative to the Late Holocene (0–2,000 years BP) (Fig. 1). Several studies suggest that such changes in Holocene ENSO characteristics were triggered by changes in Earth's orbital parameters[15,16] (Supplementary Notes 1 and 2). However, recent studies using fossil corals and mollusk shells[17,18] from the equatorial Pacific suggests that tropical Pacific climate also underwent quiescent periods throughout the Holocene that were not directly related to orbital forcing. These new studies highlight the complexity of the ENSO behaviour and the possibility that other external or internal forcings may affect the character of the ENSO response to changes in insolation.

Most state-of-the-art model simulations for the MH (with orbital insolation of 6,000 years BP) show only modest reductions in ENSO variance relative to preindustrial conditions and on the order of 5–15% (ref. 19), falling far short of capturing the full extent of Holocene ENSO changes of 30–60% represented in paleoclimate records. The reasons for this proxy-model discrepancy still remain unclear.

Most MH climate simulations are performed using the Paleoclimate Modelling Intercomparison Project Phase/Coupled Model Intercomparison Project (PMIP/CMIP) protocol, which assumes preindustrial vegetation cover and dust concentrations[20]. The PMIP/CMIP design was undertaken mainly to test model responses to changes in obliquity forcing and the seasonality of insolation. However, proxy evidence has shown that global vegetation and dust emissions differed considerably during the MH relative to the preindustrial. For example, vegetation cover expanded greatly over Sahara due to wetter climatic conditions during the MH[21–23] and Sahara dust emissions were drastically reduced as a result[24,25]. Although some modelling studies for example, (refs 26,27) include interactive vegetation schemes, they do not reproduce the magnitude of vegetation expansion suggested by proxy data, leaving most of the Sahara arid. Recent modelling efforts have shown that it is required to not only account for a more extensive greening of the Sahara, but also to include the associated reduction of dust emissions to more adequately simulate the strengthening of the West African Monsoon (WAM)[28] during the MH and its teleconnections far afield[29–31]. Interestingly, according to some proxy archives the shift in the spatio-temporal characteristics of ENSO may have occurred in approximate synchrony with the end of the Green Sahara Period ca. 3,500–5,500 yr BP[25,32–34] (Fig. 1b–d).

Here, we investigate the influence of the Sahara greening and reduced dust emissions, which are not adequately represented in the current models, and the potential teleconnection between

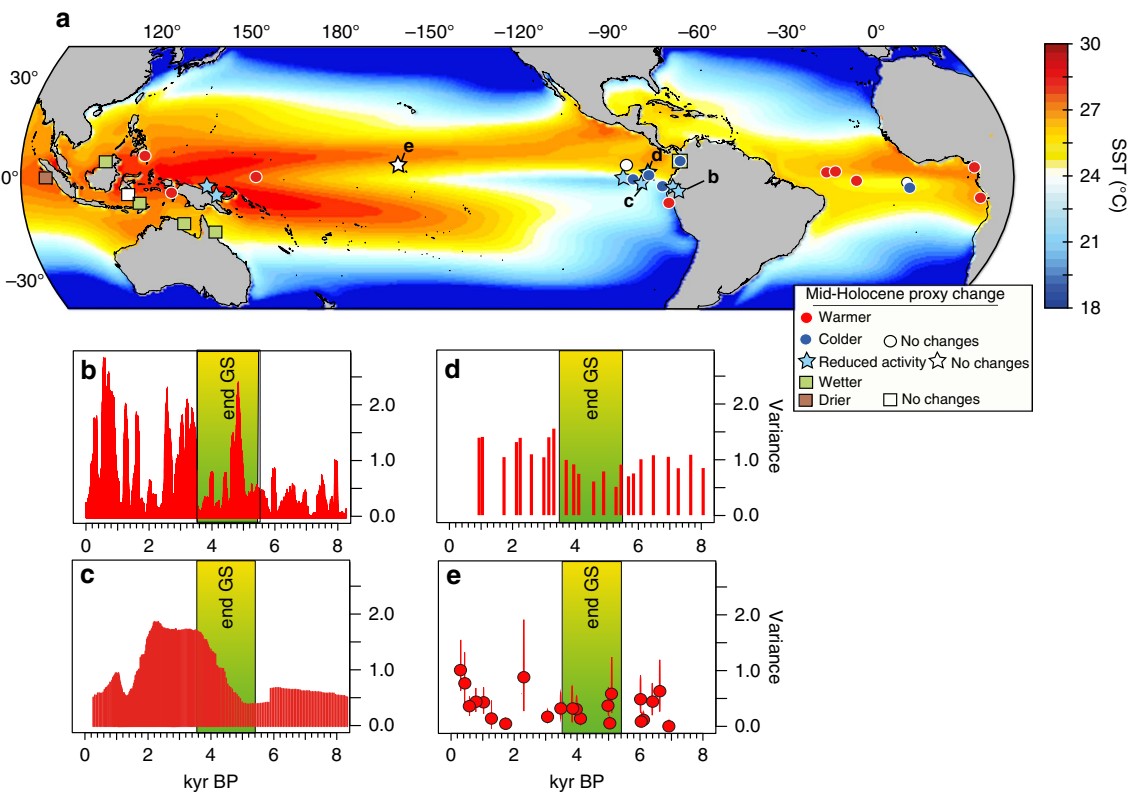

**Figure 1 | Locations of proxy archives and their recorded changes between MH and modern day climate.** (a) Locations of proxy archives with reconstructed climate changes between mid-Holocene (5,500–6,500 years BP) and late Holocene/modern climate (0–2,000 years BP) as inferred from the synthesis provided in Supplementary Tables 1 and 2. The background shading represents the climatological annual mean SST for the preindustrial climate as simulated by EC-Earth. Variance (b) of the red colour intensity of sediments[11]; (c) of sand concentration (grain concentration) in El Junco lake[8]; (d) δ[18]O of individual *G. ruber* foraminifera in core V21-30 (ref. 9); (e) and interannual variance of coral δ[18]O from the central Pacific[17,74]. The green shading indicates the end of the Green Sahara period following Shanahan *et al.*[34].

**Table 1 | Experiment boundary conditions and ENSO and ATL Niño changes.**

| Simulation | Orbital forcing | GHGs | Saharan vegetation | Saharan dust | Equatorial Atlantic ∇T | ATLN 3 s.d. | Niño 3.4 s.d. |
|---|---|---|---|---|---|---|---|
| PI | 1,850 AD | 1,850 AD | desert | PI | 0.51 °C ± 5% | 0.46 °C ± 5% | 0.51 °C ± 5% |
| ΔMH$_{PMIP}$ | 6,000 yr BP | 6,000 yr BP | As PI | As PI | − 92% | − 17% | − 10% |
| ΔMH$_{GS}$ | 6,000 yr BP | 6,000 yr BP | Shrub | As PI | − 250% | − 37% | − 23% |
| ΔMH$_{GS + RD}$ | 6,000 yr BP | 6,000 yr BP | Shrub | Reduced | − 225% | − 46% | − 25% |

Boundary conditions for each modelling experiment, the equatorial Atlantic temperature gradient in JASO (3° S–3° N 40° W–30° W; 10°–3° S 0–8° E), the standard deviations of the Atlantic Niño 3 (3° S–3° N 15°–0° W) and Niño 3.4 (5° S–5° N 120°–170° W) indexes for the preindustrial (PI) with approximate 95% confidence intervals (twice the s.e.m., see Methods section for details) and the relative changes in the MH experiments.

WAM strength and ENSO activity. Our results show that the strengthening of the WAM associated to the greening of the Sahara alters the tropical Atlantic mean state and variability, which in turn affect ENSO activity through changes in the Walker circulation. Therefore, vegetation and dust feedbacks are important players in amplifying ENSO's response to insolation forcing.

## Results

**Reduced ENSO variability enhanced by Sahara greening**. We analyse a set of idealized climate model simulations in which prescribed Saharan vegetation and dust concentrations are changed to investigate the hitherto unexplored impacts of Sahara greening on the spatio-temporal characteristics of ENSO during the MH. Our primary goal is to better understand the mechanisms behind potential variations in ENSO behaviour during the MH by investigating teleconnections between the equatorial Atlantic and Pacific that may have been influenced by changes in Saharan vegetation and dust emissions.

As a starting point, we consider the preindustrial climate (PI) as simulated by a fully coupled climate model (EC-Earth version 3.1 (ref. 35)). Another experiment (MH$_{PMIP}$) was then carried out to simulate climate of the MH using insolation and greenhouse gas boundary conditions following the PMIP3/CMIP5 protocol. We also consider two additional MH experiments in which Saharan land cover is set to shrub (MH$_{GS}$) and dust concentrations (MH$_{GS + RD}$) are reduced by as much as 80% (Table 1, Fig. 1 and Supplementary Fig. 1 in Pausata *et al.*[28]). For each experiment, we analyse a 200-year post-spin-up integration (see Methods section).

The MH$_{PMIP}$ simulation shows a significant decrease in ENSO variability of about 10% (Table 1 and Fig. 2d), which is consistent with most MH PMIP experiments[36] that only account for the insolation forcing. The timing of the peak frequency of El Niño events is also altered in our MH$_{PMIP}$ experiment relative to the PI (Fig. 3a), showing a shift in seasonality of a couple of months towards late winter/early spring, in agreement with previous studies[37]. When vegetation is imposed over the Sahara, ENSO variability is suppressed by about 23% and the length of El Niño events is reduced by about 20%. When dust concentrations are also lowered under vegetated Sahara (MH$_{GS + RD}$), the model simulates a further damping of ENSO variability to about 25% relative to the PI (Table 1 and Fig. 2f) and a nearly 50% suppression of the length of El Niño events (Fig. 3a) relative to the PI experiment. These results point to the Sahara greening as an important player in affecting ENSO activity in addition to solar insolation changes.

**Changes in Walker circulation weaken ENSO variability**. Changes in atmospheric circulation in our MH simulations affect the strength and the position of the Walker circulation (Fig. 4), causing a westward shift of the convergence and divergence maxima. In particular, over the central-eastern Pacific, an anomalous divergent flow develops during summer in all MH simulations, which is stronger in the MH$_{GS + RD}$ and MH$_{GS}$ than in the MH$_{PMIP}$ simulation (cf. Figs 4a–c and 5a,e,i). The divergent flow strengthens easterly winds over the western equatorial Pacific, but weakens them over the eastern Pacific in the MH simulations compared to PI. The weaker trades in the eastern Pacific during summer lead to decreased upwelling and a deeper thermocline in August to November (Fig. 5b,f,l) in that region. The stronger trades, instead, in the central-western part of the basin cause a shoaling of thermocline in the central Pacific in summer and the associated cold anomalies travel eastward (Kelvin wave), reaching the eastern Pacific in winter (Fig. 5b,f,l). Therefore, during winter, La Niña conditions temporary develop in the equatorial Pacific. The westward wave (Rossby wave) excited in summer by the stronger trades in the western Pacific causes deepened thermocline anomalies to the west that are subsequently reflected as a Kelvin wave (Fig. 5c,g,m). The Kelvin wave favours the eastward propagation of the deepened thermocline anomaly in the western Pacific in late summer and reaches eastern side of the basin in the following spring (Fig. 5b,f,l). Therefore, the thermocline in the eastern Pacific is eventually deepened from the earlier spring to late fall in the MH simulations compared to PI. The reduced upwelling and the deeper thermocline (reduced temperature stratification) in the eastern Pacific weaken the thermocline and upwelling positive feedback mechanisms, respectively. These are the key factors in suppressing ENSO variability through a reduction of ocean-atmosphere feedbacks (Bjerknes feedback) in the MH experiments compared to the PI simulation.

To further quantify the changes in the strength of the ocean-atmosphere coupling that affects ENSO variability in the MH experiments, we calculate the three main positive feedbacks identified in this coupling and define the Bjerknes positive feedback (BPF) index as the sum of these three components (see Methods section). The strength of ocean-atmosphere coupling is largely determined by the size of the zonal wind anomaly that is generated by sea-surface temperature anomalies and how much of its momentum is transferred as stress to the upper ocean layer. Both are then modulated by the given ocean mean state, in particular, the upwelling velocity, the stratification and zonal SST gradient.

In the MH simulations, both the ENSO amplitude (Figs 2d and 3a) and the BPF index (Fig. 3b) decrease relative to PI, in agreement with the changes in the upper ocean mean state shown in Fig. 5. The BPF index analysis confirms that the reduced ENSO variability is primarily caused by a decrease in thermocline feedback, with a moderate contribution from the upwelling feedback (Fig. 3b). The MH$_{GS}$ and MH$_{GS + RD}$ experiments show a larger decrease in BPF (17% for both) than in MH$_{PMIP}$ simulation (11%) relative to PI (Fig. 3b), which is consistent with stronger divergent flow in the central-eastern Pacific, larger thermocline anomalies and weaker upwelling displayed in summer (cf. Fig. 5b,f,l).

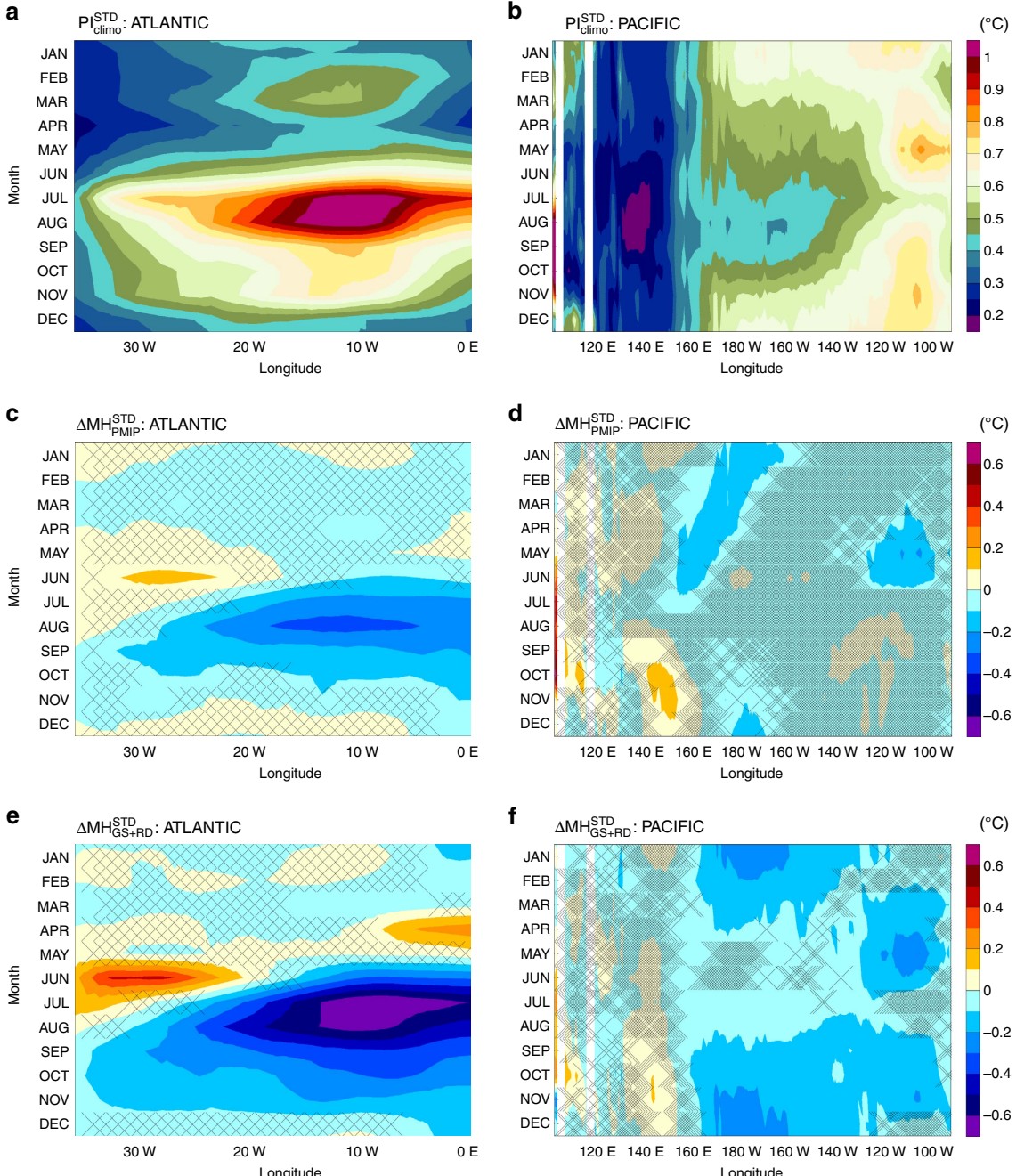

**Figure 2 | Equatorial Atlantic and Pacific longitudinal transect of the monthly average SST standard deviations for the PI simulation and the changes in the MH experiments.** (**a**) Longitudinal transect of the climatological monthly SST standard deviation for the equatorial Atlantic (3° S–3° N) and (**b**) Pacific (5° S–5° N) for each month; and (**c,d**) their changes in SST standard deviation in the $MH_{PMIP}$ and (**e,f**) $MH_{GS+RD}$ experiments for the two regions. Hatched areas reflect regions in which the changes are not significant at the 95% confidence level assessed using a two-sided t-test.

The reduction in the BPF in our $MH_{PMIP}$ simulation is identical to that found in a previous modelling study performed with the Community Climate System Model, in which the BPF has been explicitly calculated (11% decrease)[15]. Most of previous modeling studies for example, (refs 19,37–43), investigating the changes in ENSO variability in the MH did not provide a direct estimate of the changes in the BPF, but have shown ocean and atmospheric circulation changes very similar to those seen in our $MH_{PMIP}$ simulation. The atmospheric circulation changes that lead to increased ENSO stability have been associated to the orbital-induced strengthening of the South Asian monsoon[38,40].

Other studies have pointed out that the reduction in ENSO variance in the MH is owing to the changes in the tropical Pacific mean state (asymmetric west–east temperature response[16] or reduced water vapour feedback[43]) as a direct response to orbital forcing. However, our $MH_{GS+RD}$ experiment shows that the dust reduction increases shortwave radiation at the surface over the eastern equatorial Pacific (Supplementary Fig. 1c,d) and hence possibly warming the SST there (Supplementary Fig. 2). This should then decrease the east–west equatorial Pacific temperature gradient in the $MH_{GS+RD}$ relative to the $MH_{GS}$. According to the mechanism proposed in Clement et al.[16], the orbital forcing

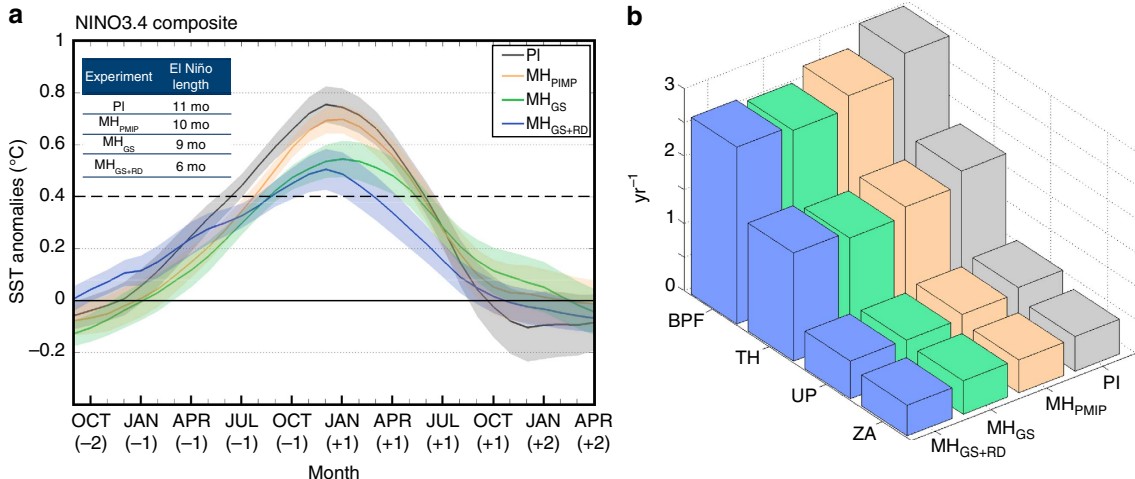

**Figure 3 | Niño 3.4 index composite and Bjerknes feedback for each experiment.** (**a**) Niño 3.4 index composite of the mean El Niño development and decay in the preindustrial (PI), MH_PMIP (only orbital changes), MH_GS (orbital and vegetation changes) and MH_GS + RD (orbital, vegetation and dust changes). The shadings show the standard error of the mean. The statistical significance of the changes has been further tested through a bootstrap technique using an extended PI simulation—575 years (Supplementary Fig. 9); (**b**) Total Bjerknes positive feedback index (BPF) and each single component (Thermocline—TH, Upwelling—UP, and Zonal Advection—ZA feedback) for each experiment. The total BPF and the relative standard error of the mean are: $3.0 \pm 0.2 \, \mathrm{yr}^{-1}$ for the PI; $2.65 \pm 0.09 \, \mathrm{yr}^{-1}$ for the MH_PMIP; and $2.5 \pm 0.2 \, \mathrm{yr}^{-1}$ for both MH_GS and MH_GS + RD experiments. The standard error of the mean has been calculated performing the BPF for sliding windows of 30 years and 10-year steps.

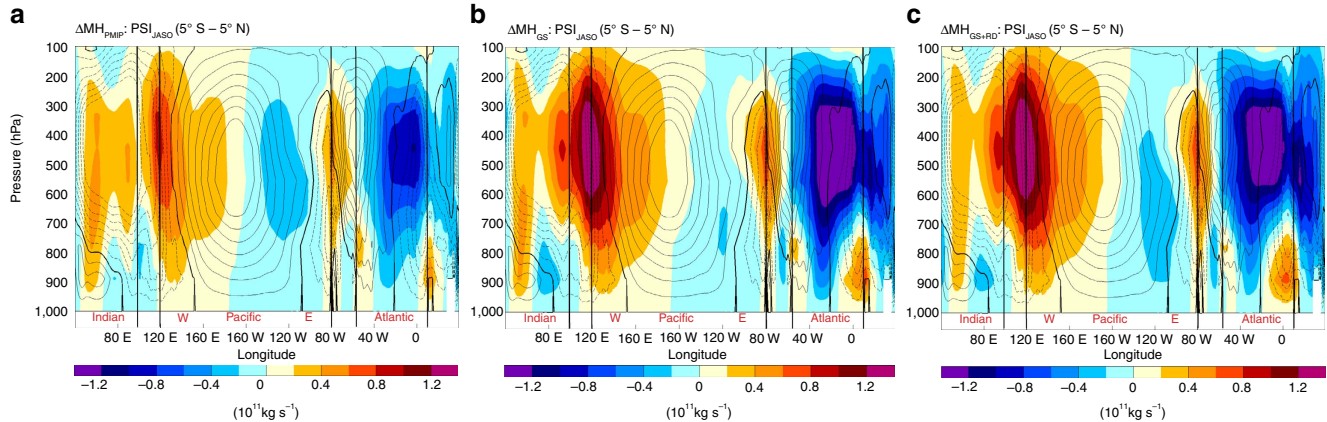

**Figure 4 | Changes in the Walker Circulation.** PI climatological zonal stream function of the Walker circulation (contours: $0.2 \times 10^{11} \, \mathrm{Kg \, s}^{-1}$ interval from $-1.4$ to $1.4 \times 10^{11} \, \mathrm{kg \, s}^{-1}$; 0 line in bold) and associated changes (shadings) in each MH experiment (**a**–**c**) relative to the PI.

should favour a stronger response—larger heating—of the atmosphere on the western than on the eastern equatorial Pacific. Consequently, the reduced east–west temperature gradient due to the dust reduction-induced warming in the eastern Pacific in the MH_GS + RD should cause an increased rather than decreased ENSO variance relative to MH_GS. Our sensitivity experiments seem then to favour a dynamical explanation more than a direct radiative effect in explaining the difference between the MH_GS + RD and MH_GS experiments: the MH_GS simulation already shows large changes in equatorial Pacific SST, where no changes in radiative forcing relative to MH_PMIP occur (Supplementary Fig. 2). A notable intensification and westward shift of the Walker circulation instead takes place in the MH_GS compared to MH_PMIP experiment (Fig. 4a,b). Our results indeed highlight that the anomalies in the Walker circulation, wind stress, thermocline, ocean current velocity and temperature stratification are somehow proportional to the strength of the WAM (Fig. 5; PI < MH_PMIP < MH_GS < MH_GS + RD), suggesting that another mechanism than previously thought may be crucial

in suppressing MH ENSO variability. This also underlines a tight link between the intensity of the climatological WAM, Walker circulation strength and position, and ENSO activity.

**Changes in equatorial Atlantic Ocean alter Walker circulation.** The main question that arises from the previous results is how the Sahara greening and the associated intensification of WAM are able to affect the MH ENSO activity. To answer this question, we turn our attention to the equatorial Atlantic. The variability of equatorial Atlantic SST is also characterized by a quasi-periodic interannual climate pattern similar to the Pacific El Niño that is termed 'Atlantic Niño'; however, the Atlantic Niño peaks during the Northern Hemisphere summer season, July to October (JASO) rather than in winter.

Our model shows an increase SST seasonal cycle with a warming taking place from late summer to late winter/early spring. The strength of the warming seems to be related to the intensity of the WAM (Supplementary Fig. 2). The JASO

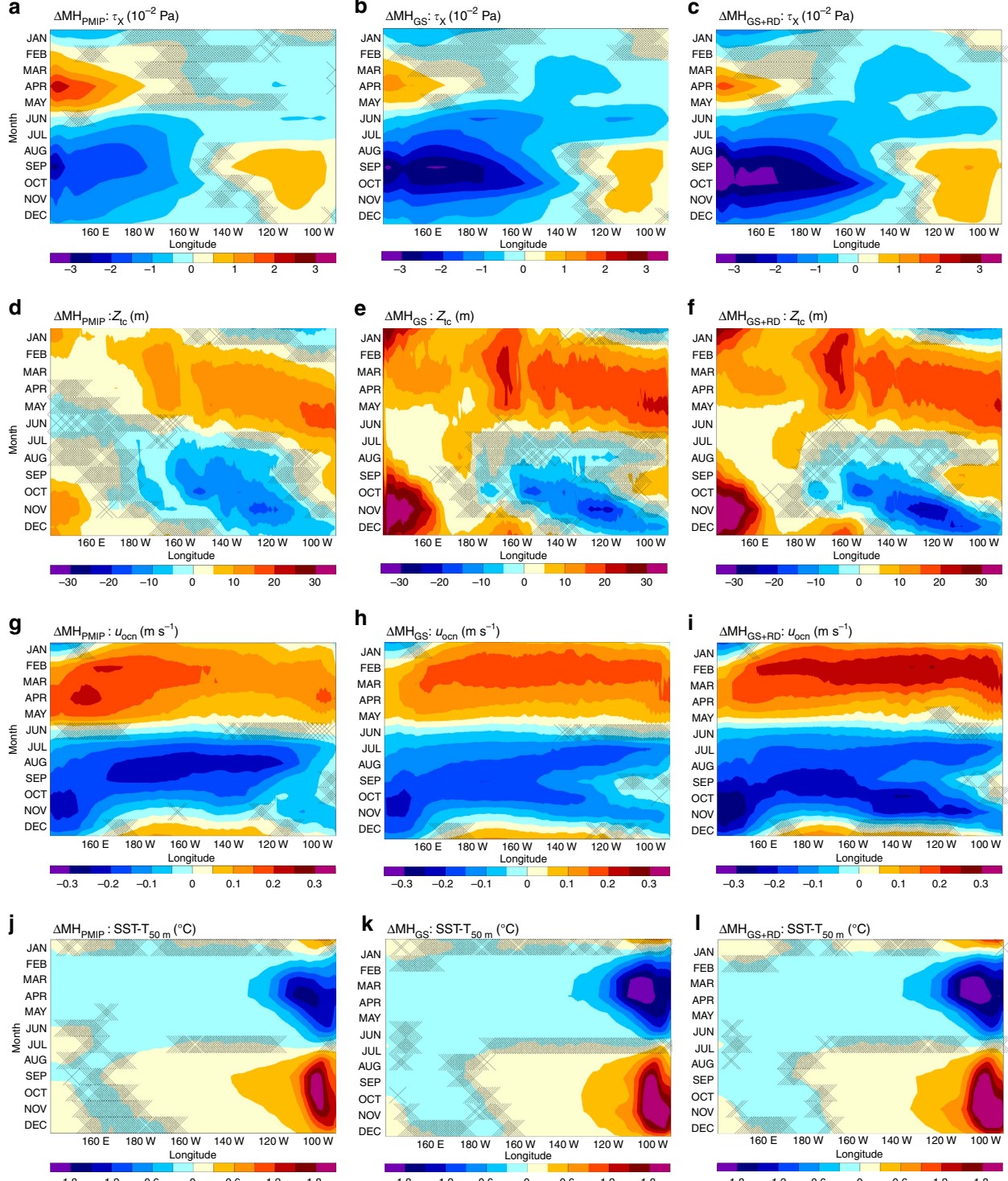

**Figure 5 | Changes in equatorial Pacific Ocean monthly mean characteristics.** (**a**–**c**) Time-longitude changes in climatological monthly eastward wind stress, (**d**–**f**) thermocline depth as captured by the 20 °C isotherm, (**g**–**i**) surface ocean current velocity, and (**j**–**l**) temperature stratification (SST—T at 50 m depth) between PI and each MH experiment. All fields are averaged between 5° S and 5° N. The hatched areas indicate regions in which the changes are not significant at 95% confidence level assessed using a two-sided t-test.

east–west temperature gradient drops of 97%, 250% and 225% in the $MH_{PMIP}$, $MH_{GS}$, and $MH_{GS+RD}$, respectively (Table 1). Accompanying the changes in Atlantic SST seasonality, a larger suppression of the Atlantic Niño variability is simulated in the MH simulations (Table 1), which is also associated with the strengthening of the WAM (Supplementary Fig. 3):

The reduction in Atlantic Niño variability goes from 17% in the $MH_{PMIP}$, to 37% in the $MH_{GS}$, to 46% in the $MH_{GS+RD}$ experiment (Table 1).

In our simulations, the intensification of the WAM in late spring and summer, along with its related wind anomalies (Supplementary Fig. 3), reduce upwelling in the Gulf of Guinea,

leading to SST mean state warming during JASO (Atlantic Niño response; Supplementary Figs 2 and 4).

The early onset of the WAM in the MH simulations (late spring, Supplementary Fig. 3) together with a northward shift of the Inter-Tropical Convergence Zone (ITCZ) consequently displace the region of maximum wind variance farther north from the equator (Supplementary Figs 5 and 6 and Supplementary Note 3). The resulting decrease in surface wind variability in the western equatorial Atlantic in late spring and summer reduces SST variability on the eastern side of the basin in JASO (Table 1 and Fig. 2), which is significantly correlated with zonal surface wind variability in the western Atlantic during spring in both observation[44] ($R = \sim 0.5$) and our model ($R = \sim 0.4$ for the PI simulation). The changes described here are larger in the $MH_{GS+RD}$ than in the $MH_{GS}$ and $MH_{PMIP}$ experiments (Fig. 2 and Supplementary Figs 5 and 6), reflecting the strength of the WAM.

The Atlantic Niño response and its decreased variability are likely able to affect the spatio-temporal characteristic of ENSO, through changes in the strength of the Walker circulation and the position of its ascending and descending branches, as shown during modern Atlantic Niño events[45,46]. The changes in the Walker circulation in our MH simulations are indeed remarkably similar to those shown during Atlantic Niño events under present day conditions (cf. Fig. 4 and Fig. 1c in Li et al.[46]). Rodríguez-Fonseca et al.[45] have shown that summer Atlantic Niños strengthen the Walker circulation with intensified descending branch over central Pacific, which favours La Niña conditions in the following winter as seen in our MH simulations. Martín-Rey et al.[47] have also shown that decreased SST variability in the equatorial Atlantic is likely to reduce ENSO amplitude. Using a coupled ocean-atmosphere model, Martín-Rey et al.[47] demonstrated that when tropical Atlantic SSTs are fixed and prescribed according to the observed monthly climatology, ENSO variability is lower than when the SSTs are interactively simulated in both basins. This further strengthens the connection between Atlantic and Pacific SST variability that is consistent with the results of our study.

To assess whether the Atlantic Niño is able to affect the Walker circulation in our model and consequently be the cause of the changes in the spatio-temporal characteristics of ENSO in the MH simulations, we perform a composite of Walker circulation anomalies associated to the Atlantic Niño phase in the PI simulation (Fig. 6). The analysis shows a net westward shift of the Walker circulation associated with positive phases of the Atlantic Niño that overall resembles the shift seen in MH simulations, in particular over western and central Pacific and western Atlantic. Over eastern equatorial Atlantic the composite displays a remarkable strengthening of the convection in the PI simulation, which is consistent with increased precipitation over the Gulf of Guinea during positive Atlantic Niños (Supplementary Fig. 7). In the MH simulations, such increase in convection over western equatorial Atlantic is much weaker and shallow (cf. Figs 4 and 6) because of the prominent shift of the rain belt well into the Sahel/Sahara region and a consequent drying over the Gulf of Guinea (Supplementary Fig. 3). The relative reduction in rainfall over the Gulf of Guinea associated with the northward expansion of the WAM and warming of eastern equatorial Atlantic SST is consistent with proxy evidence[48,49] (Fig. 1a).

As a final test to further corroborate that the Atlantic Niño response is caused by the changes in atmospheric circulation (that is, not a direct response to insolation) and is an important player for ENSO activity changes, we perform an additional simulation identical to the $MH_{GS+RD}$ but with modern day insolation forcing ($PI_{GS+RD}$). In doing so, we isolate the effect of the Sahara greening and dust reduction. Our results show

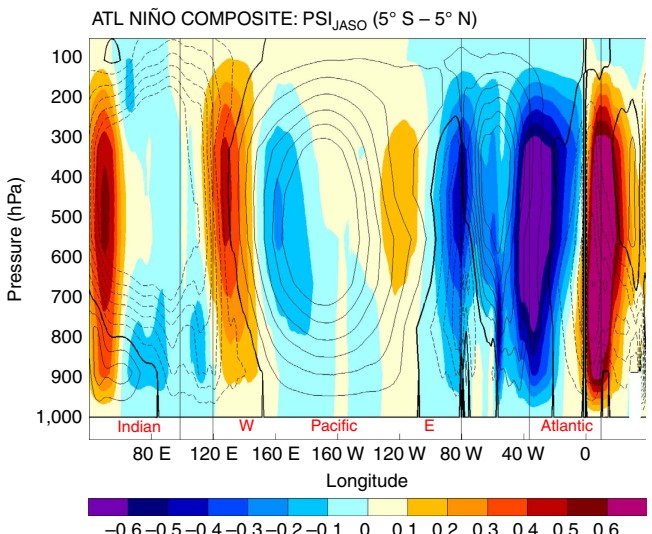

ATL NIÑO COMPOSITE: $PSI_{JASO}$ (5° S – 5° N)

**Figure 6 | Composite of Walker circulation anomalies associated with positive Atlantic Niño phases.** PI climatological zonal stream function of the Walker circulation (contours: $0.2 \times 10^{11}\,Kg\,s^{-1}$ interval from $-1.4$ to $1.4 \times 10^{11}\,Kg\,s^{-1}$; 0 line in bold) for the period June to October and its composite anomalies (positive minus negative) associated to Atlantic Niño phase. The zonal stream function composite has been calculated including the anomalies associated to Atlantic Niño and Niña events exceeding 1.5 s.d. (0.69 °C).

similar—albeit weaker—changes in the SST and precipitation pattern as those seen in the $MH_{GS+RD}$ experiments (cf. Fig. 7a,b and Supplementary Figs 2f and 3f): a stronger WAM develops over Northern Africa and Atlantic Niño conditions are present (Fig. 7a,b). The Walker circulation is shifted westward (Fig. 7c) and a La Niña response develops in winter (Supplementary Fig. 8). The equatorial Atlantic temperature gradient is reduced by 140% relative to the PI experiment. Finally, the $PI_{GS+RD}$ simulation also shows a large reduction in the standard deviation of the Atlantic Niño index of ca. 40% and of the Niño 3.4 index of ca. 13%. This additional experiment helps to disentangle the effects of Saharan vegetation and dust reduction from the effect of insolation, highlighting the relative importance of the Sahara greening in affecting the ENSO activity. Our results show that changes in the variability equatorial Atlantic SST are sources of external noise forcing that directly influence variability in the ENSO region, while changes in the mean state (Atlantic Niño anomaly) affect ENSO stability through changes in the Walker circulation. Therefore, they act in synergy in perturbing ENSO activity. Our experiments show that ENSO variability does reach its smallest values in the $MH_{GS+RD}$ when Atlantic Niño variability is minimal and the equatorial Atlantic tempera-ture gradient is strongly reduced relative to the PI. However, ENSO activity in the $MH_{GS+RD}$ is only 2% lower than in the $MH_{GS}$ experiment—albeit a notable difference in Atlantic Niño variability (9% difference, Table 1). This is likely because the equatorial Atlantic temperature gradient is instead less reduced in the $MH_{GS+RD}$ than in the $MH_{GS}$ experiment (25%), therefore partially counteracting the decrease in Atlantic Niño variability (Table 1). The increase in the $MH_{GS+RD}$ temperature gradient compared to $MH_{GS}$ is likely due to reduced dust emissions that tends to warm the western side of the equatorial Atlantic (Supplementary Fig. 3e,f).

Summarizing, the strengthening of the WAM in the MH simulations triggers an Atlantic Niño response and reduces equatorial Atlantic SST variability. These conditions over the

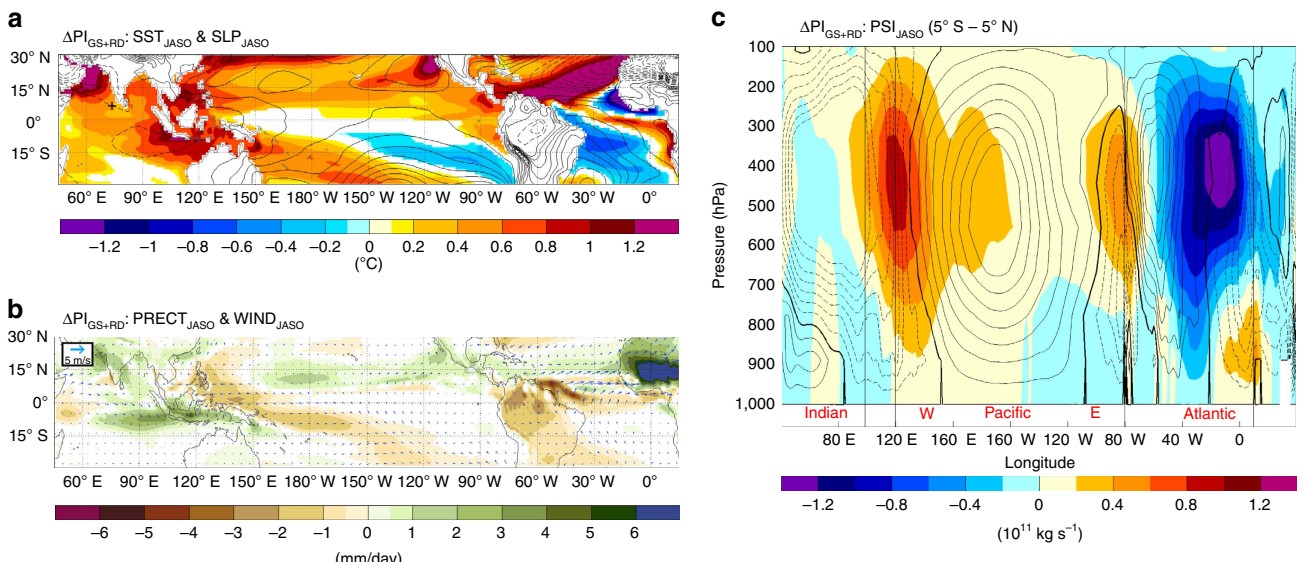

**Figure 7 | Changes in surface climate and Walker circulation for the PI$_{GS+RD}$ simulation.** (**a**) Changes in SST (shadings) and SLP (contours: 0.25 hPa interval from −3.5 to 3.5 hPa; 0 value omitted for clarity) and (**b**) precipitation (shadings) and 10 m wind (vectors) for JASO in the PI$_{GS+RD}$ simulation relative to PI. Only significant values at the 95% confidence level assessed using a two-sided *t*-test are shaded. (**c**) PI climatological zonal stream function of the Walker circulation (contours: $0.2 \times 10^{11}$ Kg s$^{-1}$ interval from −1.4 to $1.4 \times 10^{11}$ Kg s$^{-1}$; 0 line in bold) and associated changes (shadings) in the PI$_{GS+RD}$ experiment relative to the PI for JASO.

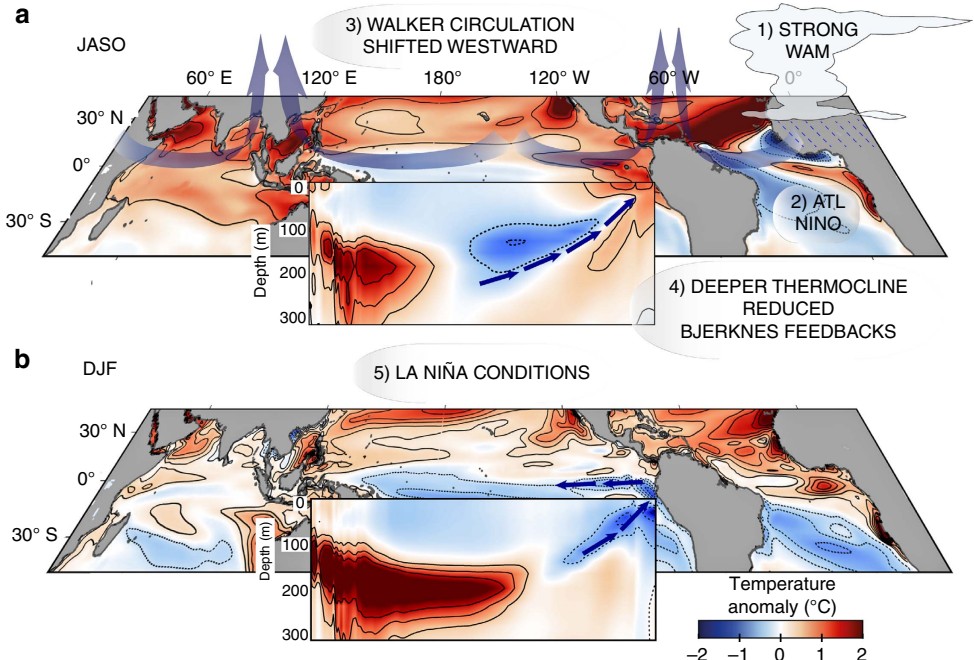

**Figure 8 | Schematic representation of the mechanisms behind the changes in ENSO variability and mean state.** (**a**) Changes in the strength of the WAM and location of the ITCZ in summer (JASO) (1) trigger an Atlantic Niño response and damp its variability (2). These conditions over the equatorial Atlantic shift the Walker circulation westward, forcing a divergent flow in the central Pacific (3). This flow cools the western Pacific and warms the eastern Pacific, with a consequent shallowing of the thermocline in the west and a deepening in the east (4) (vertical profile insert). The deepening of the thermocline and weakened upwelling in the eastern Pacific during the El Niño developing season reduces the strength of the Bjerknes feedback (4). (**b**) The thermocline anomalies in the central Pacific travel eastward reaching the eastern Pacific during winter favouring a climatological La Niña mean state in winter (5).

equatorial Atlantic strengthen and shift the ascending branch of the Walker circulation westward in boreal summer, leading to an increased divergence in the central equatorial Pacific (Figs 4, 5a,e,i and 8). The anomalous surface divergence deepens the thermocline in the east (Fig. 5b,f,l) and causes a shoaling of the

thermocline in the central part of the basin. The decreased upwelling and deepening of the thermocline in the eastern Pacific during summer—the season in which El Niño develops—reduce the strength of the Bjerknes feedback (Fig. 2b). The thermocline anomaly (shallowing) in the central Pacific in summer travels

eastward, favouring a La Niña response in winter (Fig. 8). Our results therefore suggest that the impact of orbital forcing on ENSO variability is to a large extent indirect and operates by its effect on the strength of the WAM.

**Model-proxy comparison.** Accounting for both greening of the Sahara and reduced dust emission more than doubled the magnitude of the simulated reduction in ENSO variability during the MH (6,000 years BP orbital forcing). Several key records from the western[12] and eastern[7–11] Pacific show a significant increase in variability ca. 3,500–5,500 year BP, which approximately coincides with the termination of the Green Sahara Period[34,50] (Fig. 1b–d). For example, foraminifera[9] and lake sediment[8] records from the eastern Pacific suggest damped ENSO variability during both the early and middle Holocene (Fig. 1c,d), when the Sahara was largely vegetated. Moreover, both records indicate a clear and steady increase in variability towards late-Holocene values starting around 4,500–5,000 years BP, which is in line with the reported time frame for a gradual desertification of the Sahara[25,34]. However, a recent suite of monthly- to annually resolved reconstructions based on corals and mollusk appears to be at odds with our results[14,17,18] (Fig. 1e), suggesting decreased ENSO activity during the termination of the Green Sahara period (3,500–5,500 yr BP). Such data sets, while critical to the investigation of Holocene ENSO, only cover 2,000 years of the Holocene (∼20%), with an average length of individual records of around 50 years[18]. Clearly, the basin-scale evolution of ENSO during the Holocene is still highly uncertain, and a robust data-model intercomparison of Holocene ENSO requires vastly more proxy data sets from across the equatorial Pacific. Even if such data were available, the highly idealized nature of our simulations may not capture the complex evolution of tropical Pacific variability during the Holocene. Moreover, most, if not all, paleo-ENSO proxies reflect changes in regional hydrology as well as temperature, such that the simulated changes in ENSO may be obscured by changes in the relationship of ENSO-related temperature and precipitation impacts at the various proxy sites.

In regard to changes in SST and precipitation mean state in the tropical Pacific, proxy archives seem to confirm a westward shift in the Walker circulation with increased precipitation and warmer SST in the western Pacific warm pool during the MH relative to modern climate (Fig. 1). The MH proxy data also indicate an enhanced zonal SST gradient across the equatorial Pacific, with SST cooling in the eastern part of the basin and warming in the west (Fig. 1). In contrast, our model simulates an annual net warming on both sides of the basin (Supplementary Fig. 10). Nevertheless, our modelled MH seasonal SST gradients do reveal a close match between the proxy records and the modelled SST gradient during boreal winter/early spring. It is possible that some proxy-based reconstructions of equatorial Pacific SSTs could reflect specific seasons rather than annual means, as suggested by previous studies[51–53]. For example, cooling of the warmer months and warming of the colder months in the equatorial Pacific during the MH may have shifted the preferred growing season relative to today.

In the equatorial Atlantic two proxy records from the western seaboard[48,49] (Fig. 1a), which is the region most sensitive to WAM strength changes and ocean upwelling, show warmer SST during the early and middle Holocene compared to the late Holocene. Proxy data from the central equatorial Atlantic[54] also suggest overall warmer SSTs during the Green Sahara period (Fig. 1a). Therefore, proxy archives for the MH suggest a development of La Niña conditions together with an Atlantic Niño mean state relative to today, which is consistent with our model results.

## Discussion

The changes in insolation during the early and middle Holocene caused a strengthening of the Indian and African Monsoon systems and the greening of both Asian and African deserts. The hyper-arid Sahara desert became a lush expanse of grass[21,22] and mineral dust emissions were consequently very much reduced[24,25]. Previous model studies on ENSO activity in the MH have shown a reduction of its variability around 10%, but they have focused on changes in orbital forcing alone for example[15,37,38,43], ignoring modifications in vegetation and dust conditions. Sediment and foraminifera records[7–11] have instead suggested a larger suppression of ENSO activity (30–60%) during the MH[12,14,15]. In the attempt to explain the underlying causes of such changes in ENSO behaviour, previous studies have pointed either to the direct response to the orbital forcing in altering the tropical Pacific mean state[16] or to enhanced trades winds associated to stronger Asian monsoon[38,40,42]. Here, we propose that the changes in the WAM are instead the crucial factor for altering the Walker circulation, through changes in equatorial Atlantic SSTs (Atlantic Niño) (Fig. 8). We demonstrate the importance of Saharan vegetation and the associated decrease in airborne dust concentration in amplifying the mechanisms by which the ENSO variability is reduced in the MH_PMIP simulation. The Saharan vegetation and dust reduction act as additional players in enhancing the WAM northward expansion and its associated feedbacks.

In summary, our study suggests that ENSO variability is sensitive to the strength of the WAM—a finding with implications for the study of both past and future ENSO. Recent modeling studies indeed suggest a strengthening of the WAM and less dusty Sahara in a future warmer climate[55–57], which may in turn affect ENSO activity as shown in our work. Hence, to better represent past and future ENSO response, modeling studies should include changes in Saharan vegetation and dust emissions. More continuous and high-resolution proxy records from both the Pacific and Atlantic Ocean that are able to capture both SST mean state and variance are critically needed. These proxy archives will allow for the quantification of relative contributions of forced versus internal variabiilty in ENSO and better elucidate the climatic changes that occurred in the tropics throughout the Holocene. Our study suggests that improvements in the simulation of vegetation and dust feedbacks are of paramount importance in modeling future tropical climate changes that will strongly affect the livelihoods of millions of people.

## Methods

**Model description.** We used version 3 of the climate model EC-Earth[35] to design numerical simulations of the preindustrial and the mid-holocene climates. The atmospheric model is based on the Integrated Forecast System (IFS cycle 36r4) developed by the European Centre for Medium-range Weather Forecasts, including the H-TESSEL land model. The simulation is run at T159 horizontal spectral resolution (roughly $1.125° \times 1.125°$) with 62 vertical levels.

The ocean model is version 3.3.1 of the Ocean General Circulation Model—NEMO[58]. It solves the primitive equations discretized on a curvilinear horizontal mesh with a horizontal resolution of about $1° \times 1°$ and 46 vertical levels. At the surface, the model is coupled every model hour with the Louvain-la-Neuve Ice Model—LIM3 (ref. 59) that as the same horizontal resolution as NEMO. The coupling between the NEMO-LIM system and the atmospheric model IFS was carried out with the coupler OASIS3 (ref. 60).

EC-Earth has been extensively used for simulating past, historical and future climate contributing to the Fifth Assessment Report of the Intergovernmental Panel on Climate Change[61] and to the Paleoclimate Modeling Intercomparison[62]. EC-Earth has shown good skills in representing monsoonal precipitation both temporally and spatially in present day climate[28,62]. The tropical variability over the Pacific is well capture, however, somewhat underestimate compared to observations (Table 1 and Hazeleger et al.[35]). The typical ENSO pattern is present in EC-Earth, but it extends too far to the west compared to observation and the ENSO return time is around ∼7 years (3–7 years in observations). The atmospheric aspects of ENSO are well captured in EC-Earth and the Walker circulation is correctly represented (cf. contours in Fig. 4 and Fig. 2a in

Bayr et al.[63]. EC-Earth is unable to correctly capture the SST gradient across the equatorial Atlantic, problem common to many climate models. Nevertheless, the model is able to correctly capture Atlantic Niño variability in terms of seasonality and variability (0.46 versus ∼0.4 in observations, Fig. 1b in Nnamchi et al.[44]). The model is also able to simulate the correlation between the strength of surface wind in spring and summer ATL Niño 3 variability (∼0.4 versus 0.5 in observations, see Fig. 1a in Nnamchi et al.[44]). For a detailed description of EC-Earth performance refer to Hazeleger et al.[35,64].

Boundary conditions for the mid-Holocene control (MH$_{PMIP}$), except for the orbital forcing and greenhouse gases, were set at preindustrial values according to the PMIP/CMIP5 protocol[20]. This includes land surface, aerosols, ice sheets, topography and coastlines. The orbital forcing was set at 6,000 years BP values and computed internally using the method of Berger[65]. Differences in the Earth's orbit in the MH enhanced the amplitude of the seasonal cycle in Northern Hemisphere insolation by ∼5% compared to present day values. For the greenhouse gases we changed methane concentration from 760 ppmv PI value to 650 ppmv for MH according to PMIP3/CMIP5 protocol, and kept $CO_2$ and other greenhouse gases the same as PI. Vegetation cover and properties, and dust concentrations are prescribed. The dust distribution used in this study and in Pausata et al.[28] was taken from the Community Atmosphere Model (CAM)[66], which is used in the Coupled Model Intercomparison Project (CMIP) phase 5. A second set of experiments is carried out in which the vegetation type over the Sahara domain (11°–33° N and 15° W–35° E) is set to shrub (MH$_{GS}$) and the PI dust amount is also reduced by up to 80% (Fig. 1 and Supplementary Fig. 1 in Pausata et al.[28]), based on recent estimates of Saharan dust flux reduction during the MH[24,25] (MH$_{GS+RD}$). The vegetation change corresponds to a reduction in the surface albedo from 0.3 to 0.15 and an increase in the leaf area index from 0.2 to 2.6 (mainly desert and shrub, respectively, Table 1 in ref. 28). The dust reduction leads to a decrease in the dust aerosol optical depth of almost 60% and in the global total AOD of 0.02 (Fig. 1 in Pausata et al.[28]). The 80% dust reduction was applied over a broad area around the Sahara desert from the nearby Atlantic Ocean to the Middle East and throughout the troposphere (up to 150 hPa). A smoothing filter was used to avoid abrupt transitions in dust concentrations (Supplementary Fig. 1b,d,f in Pausata et al.[28]). Above 150 hPa the dust reduction was more evenly applied due to the fact that aerosol particles are uniformly distributed at those elevations. The change in dust concentration and vegetation cover are not meant to provide a faithful representation of the MH conditions over the Sahara and nearby regions, since no accurate vegetation reconstruction is available at the moment. They have instead been designed to more easily disentangle the effects of land surface cover and dust loading on atmospheric circulation. The details of the boundary conditions for the sensitivity experiments are listed in Table 1. Initial conditions for the MH experiments were taken from a 700-year preindustrial spin-up run, and the simulations were then run for about 300/400 years. The climate reaches quasi-equilibrium after 100–200 years, depending on the experiment. In this paper, we focus on the equilibrium responses, and only the last 200 years of each sensitivity experiment are analysed.

**NIÑO3.4 and ATL NIÑO 3 indices.** The ENSO index used in this study consists of monthly mean SST anomalies spatially averaged over the Nino3.4 region (5° N–5° S and 170° W–120° W). A 5-mo running mean is applied to damp uncoupled intra-seasonal variations in SST. El Niño events are defined as the periods during which the 5-mo running mean of the SST index anomaly is greater than +0.4 °C for at least six consecutive months. Changes in the ENSO variability are measured as changes in the SST SD in the Niño 3.4 area. The ATL Niño 3 index is defined in the following domain 3° S–3°N and 15–0° W.

**Significance of ENSO and ATL NIÑO changes.** To quantify the changes in ENSO activity, we have used the standard deviation because Brown et al.[19] found it to be preferable to event frequency or size as event-based measures are highly dependent on the choice of threshold and may be unreliable for a small number of events. To test whether the changes in ENSO and Atlantic Niño standard deviation and seasonal cycle in the MH experiments were significant we have extended the PI simulation to 575 years. Hence, through a bootstrap technique, we have randomly extracted 100 random sub-samples of continuous 200 years from the original pool of 575 years. We then calculate the standard deviation for each sub-sample: The standard deviation of bootstrapped quantities is by definition an empirical estimate of the standard error of the mean[67] and is what we have used to determine whether the changes in the standard deviations of Niño 3.4 and ATL Niño 3 indexes in the MH experiments were significantly different from the PI reference simulation. We also make the Niño 3.4 index composite as shown in Fig. 3 for each of the 100 sub-samples of the PI experiment to explore the maximum range of variability (Supplementary Fig. 9 and Supplementary Note 4). The analysis shows that the changes in the Niño 3.4 seasonal cycle in the MH$_{GS+RD}$ are significantly different from the PI experiment (>99th percentile).

**Evaluation of the Atlantic/Pacific teleconnection in our model.** Given the teleconnection between the tropical Atlantic and Pacific presented in this study, we deem relevant to test our model performances in capturing the ENSO/Atlantic Niño/WAM mode of variability. Previous studies[68,69] have shown that although

the models are able to reproduce the observe modes of variability, the impact of tropical SSTs on the WAM precipitation is generally overestimated. Here, we use the maximum covariance analysis (MCA) to evaluate this teleconnection. The MCA can be considered as a generalization of the principal component analysis. The MCA analysis looks for patterns in two space-time data sets (SST and precipitation in our case), which explain a maximum fraction of the covariance between them. The MCA provides two sets of singular vectors and a set of singular values that are associated with each pair of vectors. Each pair of vectors represents a fraction of the squared covariance between the two variables (SCF: squared covariance fraction). The expansion coefficients for each variable are calculated by projecting the respective data field onto the respective singular vector. The correlation value (Corr.) between the expansion coefficients of the two variables indicates how strongly related the coupled patterns are.

We compared the MCA performed using the output of the PI simulation to MCA calculated based on the precipitation and SST data set provided respectively by the NOAA/OAR/ESRL PSD (Global Precipitation Climatology Project (GPCP) Combined Precipitation Data Set Version 2.2: http://www.esrl.noaa.gov/psd/) and the Hadley centre (HadISST[70]) for the period 1979–2013.

The analysis shows that overall the model is able to reproduce the leading mode of covariance between the SST and precipitation pattern in North Africa: Atlantic Niño and La Niña conditions are associated with increased precipitation over the Gulf of Guinea. While our model shows a similar squared covariance fraction (62 versus 67% in our model) for the leading mode of covariance as well as correlation between SST and precipitation over northern Africa (0.51 versus 0.61), the precipitation anomaly is too strong, as in most climate models, and shifted southward compared to the reanalysis data sets (Supplementary Fig. 7b,d). In our model, the SST pattern shows weaker anomalies in the tropical Pacific and stronger in the Atlantic relative to the reanalysis. The southward shift in our model is likely due to the WAM dry bias: the WAM's northernmost extent is located at 14.0° N about 250 km too far south compared to observation (Supplementary Fig. 6 and relative discussion in Pausata et al.[28]).

The strong correlation between the expansion coefficients associated to SST and precipitation time series has been interpreted as a modulation of the monsoon activity by tropical Pacific SST[68,69]. However, the modulation of the monsoon activity in northern Africa is also caused by the Atlantic Niño phase, which in turn can affect the tropical Pacific SST.

In our model, the SST pattern shows weaker anomalies in the tropical Pacific and stronger in the Atlantic relative to the reanalysis. This can be seen as a stronger influence of the tropical Pacific on the Atlantic SST or a more feeble impact of the Atlantic Niño on ENSO in our model compared to the reanalysis. Therefore, it may well be that the effects on ENSO activity during the MH associated to the Atlantic Niño response may have been larger than those simulated. However, more in-depth analyses are needed to disentangle this aspect, which is beyond the scope of this study.

**Bjerknes positive feedback index.** The BPF index consists of the three main positive feedbacks identified in the atmosphere-ocean coupling: the zonal advection feedback (ZA), the Ekman local upwelling (UP) feedback, and the thermocline (TH) feedback[71]. The BPF index can then be express as:

$$\mathrm{BPF} = \underbrace{\mu_a\beta_u\langle -\overline{T_x}\rangle}_{\mathrm{ZA}} + \underbrace{\mu_a\beta_w\langle -\overline{T_z}\rangle}_{\mathrm{UP}} + \underbrace{\mu_a\beta_h\left\langle\frac{\bar{w}}{H_1}\right\rangle a_h}_{\mathrm{TH}}, \qquad (1)$$

where the overbar denotes the annual mean climatology, $\langle\overline{T_x}\rangle = \langle\frac{\partial\bar{T}}{\partial x}\rangle$ and $\langle\overline{T_z}\rangle = \langle\frac{\partial\bar{T}}{\partial z}\rangle$, $T$ is sea temperature anomaly and the angle brackets indicate averaged quantities over the equatorial central and eastern Pacific Ocean (5° S–5° N, 180° E–80° W). $H_1$ is the mixed layer depth and w the ocean vertical velocity. The coefficients in equation (1) can be calculated using least-square regression method (for more details see[15,72,73]). Each of the three dynamical positive feedbacks is proportional to the atmospheric response sensitivity ($\mu_a$), and each is further proportional to its own oceanic response sensitivities ($\beta$). All three feedbacks are the product of the background state ($d_x\bar{T}$, $d_z\bar{T}$ and $\bar{w}$), the atmospheric response sensitivity (or surface wind stress sensitivity) to SST ($\mu_a$), and the oceanic response sensitivity to equatorial surface wind stress ($\beta_u$, $\beta_w$ and $\beta_h$), reflecting the critical role of each element in the generation of the feedback. In addition, the TH feedback is proportional to entrainment temperature response sensitivity to local thermocline depth ($a_h$). The BPF index represents the total positive feedback strength and is an indicator of the atmospheric-ocean coupling.

**Data availability.** The data sets generated and analysed during the current study are available from the corresponding author on reasonable request.

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

## Acknowledgements

The authors would like to thank Marco Gaetani and Gabriele Messori for helpful discussions. FSRP acknowledges funding from the Swedish Research Council (FORMAS) as part of the Joint Programming Initiative on Climate and the Belmont Forum for the project 'Palaeo-constraints on Monsoon Evolution and Dynamics (PACMEDY)'. Q.Z. acknowledges funding from Swedish Research Council VR for the Swedish-French project GIWA. L.C. acknowledges funding through the iNcREASE project. E.M.N. acknowledges funding through the LOEWE funding program (Landes-Offensive zur Entwicklung wissenschaftlich-Ökonomischer Exzellenz) of Hesse's Ministry of Higher Education, Research, and the Arts. J.C.S. acknowledges funding by the United States National Science Foundation. The climate model simulations and analysis were performed on resources provided by the Swedish National Infrastructure for Computing (SNIC) at NSC and Cray XC30 HPC systems at ECMWF. Z.Li and Z.Lu acknowledge the support from China National Science Foundation 41630527 and NSF P2C2.

## Author contributions

F.S.R.P. and Q.Z. conceived the study; F.S.R.P. designed the experiments, analysed the data and wrote the manuscript; Q.Z. performed the coupled model simulations; Q.Z., Z.Lu. and L.C. analysed the data F.M. and J.C.S. compiled, analyzed and interpreted the proxy data with contribution from F.S.R.P., E.M.N., and K.M.C. authors contributed to the discussion of the results and the writing of the manuscript.

## Additional information

**Competing interests:** The authors declare no competing financial interests.

