## [Peer Review File · Nature Communications]

Editorial Note: this manuscript has been previously reviewed at another journal that is not operating a transparent peer review scheme. This document only contains reviewer comments and rebuttal letters for versions considered at Nature Communications

Reviewers' comments:

Reviewer #3 (Remarks to the Author):

This reads much like a "this is what ENSO does in my model" paper. In my opinion, there are many of these manuscripts in the literature at present, which I think means that it is not novel. The links to the paleo work are only done in attempt to validate the model, but observations are much better suited to this given the discrepancy amongst paleo sources. The only novel aspect is the links to the Sahel region, but this is still the results of just one model. The authors go into a bit of detail to try and explain how and why the changes impact ENSO, but all I keep thinking is would I find a similar result if I did the same experiment in a different model. As shown in the results of Bellenger et al. 2014 (in climate dynamics), while most models do a reasonable job of representing ENSO, there is a large amount of underlying error compensation occurring which raises the question are they getting it right for the right reasons.

In my opinion, the authors should aim for a longer format journal to better explain the details of the atmospheric teleconnections and how this leads to changes in ENSO.

Reviewer #4 (Remarks to the Author):

This manuscript presents the results of sensitivity experiments of the mid Holocene climate showing that, with the model used, the damping of ENSO variability is larger when Sahara is vegetated and dust are taken into account. It suggests that this is due to the damping of tropical Atlantic variability. The paper also includes a comparison with various proxy records for a broader discussion on ENSO in the Holocene. I accepted to review this paper as an additional external reviewer with the request to check if the questions raised by one of the previous reviewers were properly addressed in the revision and to provide my own view on the modeling study and the conclusion. I found this study clearly written and interesting.

The strength of the paper is that it raises the possibility that enhanced mid-Holocene West African monsoon would have contributed to damp ENSO variability, which was due to a combination of insolation, vegetation and dust forcings. However, the paper provides the feeling that the authors propose a new data synthesis that can help to assess this teleconnection, which is not fully done in the manuscript. In this respect I share some of the concerns of reviewer 2, and even though the revision addresses most of the comments about the messages, it doesn't fully address the questions on the data reconstructions and on model-data comparison. I would also recommend that the analyses, and associated figures, focus on the particular role of vegetation and dust and that the part on the response to insolation is reduced. These two comments are expanded below. Reading the reviewer comments and the responses on the model-data comparison I realized that the authors slightly misunderstood the major reviewer comment or criticisms. They provide lots of arguments in the response to the editor to defend their study, and these arguments are not used in the text to try to reconcile or discuss the different records. Also the discussion on the proxy mostly concern ENSO variability when they also use several reconstructions of the mean climate. Proxy records are mentioned in Figure 1, but not fully analyzed, and the text doesn't tell what is the new conclusion that can be drawn from this figure. Is it a new overview picture supposed to bring new information compared to previous data synthesis over this region?. Is it only a figure to show where the different records are located? It is also not clear to me why table 1 only includes coral records from Tudhope et al. (2001) and do not take on board other coral records, such as those of Cobb who is listed as a co-author and also contributed to Emile-Geay et al. 2016 data synthesis. The authors should also decide if they include or not a rigorous model-data comparison in this manuscript, which would require a more complete discussion of available records and of what would be needed to go one step further so as to properly connect the different regions and discuss the linkages between interannual and annual or seasonal changes. As far as I understand reviewer 2 comment, the point is not to tell that mollusk or corals are better than the other sources, but to put all the sources together with their relative uncertainties. If they do it, they should tell what

figure 1 brings compared to previous figures like this (I.E Braconnot et al. 2012), or to previous mid-Holocene or Holocene reconstructions (I.E the different sources contributing to Emile-Geay et al. 2016). Since the discussion is about the linkages between the WAM and ENSO, it would be appropriate to also include the African and Atlantic sector in this figure and in the discussion. If they do not do this, they should then only refer to the literature and include this figure in the additional material. They should consider nevertheless to have a figure comparing their model results to the different records they consider in the different places, even though we know that the sensitivity experiment represent an unrealistic extreme case of the enhanced mid-Holocene African monsoon.

I would also recommend that the text focuses on the sensitivity experiments and bring more information on the role of vegetation and dust. Most of the mechanisms discussed in the text to explain the ENSO response to insolation forcing have already been extensively discussed in previous studies (I.E Brown et al, 2008, Zheng et al. 2008, Luan et al. 2012, An and Choi 2014, Karamperidou 2015, Roberts et al., 2015, and more recently Saint-Lu et al. 2016 for the energetics). The authors should refer to these manuscripts to tell how their mid-Holocene simulation compares to previous studies. In particular, the multi-model analysis of An and Choi 2014 provides an estimate of the feedback terms for mid-Holocene simulations. A rapid comparison with what has already been done would thus be welcome. Figure 4 as it stands brings nothing new compared to these previous studies. It should also show similar plots to highlight what is the additional effect of vegetation and dust compared to insolation alone. Also the feedback estimates should come with error bars, since some of the models have very large internal noise (cf Emile-Geay et al. 2016 for it) or diversity in ENSO characteristics (Wittenberg et al. 2009). Additional analyses could be provided on the mechanisms by which ENSO variability could be linked to WAP. The argument is based on the fact that both ENSO variability and tropical Atlantic variability are damped, using results from a published sensitivity experiment on present day climate. Part of it could be better documented from their simulations. The inclusion of the radiative effect of dust seems to partly counteract the effect of vegetation. Is there something to learn from the differences between these two simulations? What about the change in the west Pacific seen in the addition material figures? Is there a change in stratification related to the west Pacific wind (the additional material suggest that there isn't), or could those winds counteract the generation or propagation of Kelvin waves? There are differences between the simulation with vegetation and with dust that could also be used to point out the important factors.

Minor points. The text is very careful not overselling too much the results and recognizes that the sensitivity experiments represent extreme possible limit of W African vegetation and dust effect. A few words on the limits of the models in representing WAM/ENSO teleconnection should be mentioned, as well as previous finding of changes of this teleconnection in a different climate. Studies with previous model generation highlighted that ENSO/WAM teleconnection was too strong in most models (Jolly et al. 2007, Zhao et al. 2007). Could this affect the teleconnection mentioned here?. The method or the additional material should provide addition information on the simulations and the way the model is set up for the different simulations. Are the authors sure that the model is exactly the same (I.E same feedback included) when they impose the vegetation and run with the dust model? How is dust considered in the different simulations, including the pre-industrial control?

Response to Reviewer #4:

1) The strength of the paper is that it raises the possibility that enhanced mid-Holocene West African monsoon would have contributed to damp ENSO variability, which was due to a combination of insolation, vegetation and dust forcings. However, the paper provides the feeling that the authors propose a new data synthesis that can help to assess this teleconnection, which is not fully done in the manuscript. In this respect I share some of the concerns of reviewer 2, and even though the revision addresses most of the comments about the messages, it doesn't fully address the questions on the data reconstructions and on model-data comparison. I would also recommend that the analyses, and associated figures, focus on the particular role of vegetation and dust and that the part on the response to insolation is reduced. These two comments are expanded below.

We did not mean to do a new data synthesis, but focus on the modeling result regarding the new mechanisms connecting the WAM to Atlantic Niño and hence to ENSO. We briefly compare our model results with the available proxy data. Figure 1 was meant for the reader to have at hand the overall picture of inferred changes in mean climate and variability based on the available proxy data. Our study is not a comprehensive global review of ENSO proxy records, which though much-needed, would be a large project of its own and far beyond the scope of this modeling paper. Nevertheless, we have followed the reviewer comments to discuss more in detail the available data and their shortcomings. In the revised version of the manuscript, we have also modified figure 1 to include the Atlantic Ocean proxy archives, a discussion of the data shown in the figure and expand the analysis of the role of vegetation and dust. See detail in the answers to specific comments below. If the reviewer and/or the editor prefers, we will be happy to move the model-proxy data comparison to the supplementary material, since our manuscript aim at understanding the dynamical mechanisms that can potentially affect ENSO.

2) Reading the reviewer comments and the responses on the model-data comparison I realized that the authors slightly misunderstood the major reviewer comment or critics. They provide lots of arguments in the response to the editor to defend their study, and these arguments are not used in the text to try to reconcile or discuss the different records. Also the discussion on

the proxy mostly concern ENSO variability when they also use several reconstructions of the mean climate. Proxy records are mentioned in Figure 1, but not fully analyzed, and the text doesn't tell what is the new conclusion that can be drawn from this figure. Is it a new overview picture supposed to bring new information compared to previous data synthesis over this region? Is it only a figure to show where the different records are located?

Thanks for pointing this out. We have now extended the model-proxy comparison to better illustrate the proxy records presented in figure 1, also highlighting the shortcomings of the different reconstructions and discussing the mean state changes both in Pacific and the Atlantic. The model-proxy comparison section now reads:

“Accounting for both greening of the Sahara and reduced dust emission more than doubled the magnitude of the simulated reduction in ENSO variability during the MH (6,000 years BP orbital forcing). Several key records from the western¹² and eastern⁷⁻¹¹ Pacific show a significant increase in variability ca. 3,500-5,500 yr BP, which approximately coincides with the termination of the Green Sahara Period^{20,50} (Fig. 1b-d). For example, foraminifera⁹ and lake sediment⁸ records from the eastern Pacific suggest damped ENSO variability during both the early and middle Holocene (Fig. 1c, d), when the Sahara was largely vegetated. Moreover, both records indicate a clear and steady increase in variability towards late-Holocene values starting around 4,500-5,000 years BP, which is in line with the reported time frame for a gradual desertification of the Sahara^{20,27}. Finally, the foraminifera record⁹ also shows higher ENSO variability during the LGM, a period characterized by a drier Sahara-Sahel (weak WAM)⁵¹, lending further support to a potential relationship between ENSO and WAM. This temporal association of changes in ENSO behaviour with the strength of the WAM seems to argue in favour of an active trans-basin teleconnection between the two regions. However, a recent suite of annually-resolved reconstructions based on corals and mollusk appears to be at odds with our results^{14,17,18} (Fig. 1e), suggesting a decreasing trend in ENSO activity during the termination of the Green Sahara period (3,500-5,500 yr BP). The observed trend during this period in those records is thus opposite to what should be expected by our results. However, the idealized nature of our simulation does not allow to directly comparing our results with the proxy archives and to fully reproduce the complex evolution of tropical Pacific variability during the Holocene, but rather explore the underline mechanisms and some of the possible influences that are able to affect ENSO – which is the aim of this study. Moreover, there are several factors that preclude a rigorous data-model evaluation at this point in time. First, corals and mollusk proxy archives, which are among the best-suited proxies for recording ENSO variability given their high-temporal resolution (annual), are compilations of short records and there are still too few ENSO-resolving datasets from across the tropical Pacific to determine any change in the spatial character of ENSO during the MH. For example, the most recent synthesis¹⁸ of all corals and mollusk data of the tropical Pacific comprising the last 10,000 years covers a total of only 2,000 years, with an average length of individual records of around 50 years¹⁸. As ENSO activity is non-stationary, inferring ENSO variability over such short temporal windows and discontinuous temporal coverage can lead to a broad range of estimates⁵² with variations up to $\pm 50\%$ ¹⁸. Second, as most, if not all, paleo-ENSO proxies reflect changes in regional hydrology as well as temperature, the simulated changes in

ENSO may be obscured by changes in the relationship of ENSO-related temperature and precipitation impacts that are difficult to distinguish in MH proxy records. Lastly, the ENSO behaviour may have been affected by many external and/or internal forcings, acting in synergy with the changes in solar insolation, vegetation and dust emissions.

In regard to changes in the SST and precipitation mean state in the tropical Pacific, proxy archives seem to confirm a tendency towards more La Niña conditions during the MH (Fig. 1a and Table S1), which agrees with our results at least for the winter (Figs. S3 and S10, see also discussion in Supplementary Information). Proxy records show an increased east – west temperature in the tropical Pacific and increased precipitation in the western Pacific warm pool, which is consistent with the westward shift in the Walker circulation, precipitation and wind anomalies displayed in our MH simulations (cf. Fig. 1 and S2). In the equatorial Atlantic two proxy records from the western seaboard^{48,49}(Fig. 1a), which is the region most sensitive to WAM strength changes and ocean upwelling, show warmer SST during the early and middle Holocene compared to the late Holocene. Proxy data from central equatorial Atlantic⁵³ also suggest overall warmer SSTs during the Green Sahara period (Fig. 1a). Therefore, proxy archives for the MH suggest a development of La Niña conditions together with an Atlantic Niño mean state, which is consistent with our model results.”

Figure 1 has also been extended to the Atlantic. Only proxies for mean state change are currently available in the equatorial Atlantic Ocean.

3) It is also not clear to me why table 1 only includes coral records from Tudhope et al. (2001) and do not take on board other coral records, such as those of Cobb who is listed as a co-author and also contributed to Emile-Geay et al. 2016 data synthesis.

In the revised version we had submitted in late November, the coral records from the central Pacific (Cobb et al, 2013) were included both in figure 1 (white star) and table S1. We also now included a panel showing the combination of two proxy from central Pacific (Cobb et al., 2003, 2013). Finally, now we also present an additional coral data from the western Pacific (McGregor & Gagan, 2004).

4) The authors should also decide if they include or not a rigorous model-data comparison in this manuscript, which would require a more complete discussion of available records and of what would be needed to go one step further so as to properly connect the different regions and discuss the linkages between interannual and annual or seasonal changes. As far as I understand reviewer 2 comment, the point is not to tell that mollusk or corals are better than the other sources, but to put all the sources together with their relative uncertainties. If they do it, they should tell what figure 1 brings compared to previous figures like this (I.E Braconnot et al. 2012), or to previous mid-Holocene or Holocene reconstructions (I.E the different sources contributing to Emile-Geay et al. 2016). Since the discussion is about the linkages between the WAM and ENSO, it would be appropriate to also include the African and Atlantic sector in this figure and in the discussion. If they do not do this, they should then only refer to the literature and include

this figure in the additional material. They should consider nevertheless to have a figure comparing their model results to the different records they consider in the different places, even though we know that the sensitivity experiment represent an unrealistic extreme case of the enhanced mid-Holocene African monsoon.

As mentioned above an in depth synthesis of the available proxy data is beyond the scope of this manuscript. As reported above, we have now expanded the discussion and included the Atlantic sector into figure 1, which however do not show any new proxy data but it's meant to be as a helpful overview for the readers. The proxies presented in figure 1 are now discussed in more detail.

In the concluding paragraph we have also mentioned what will be needed to go one step further in our understanding of the African Humid Period and the WAM/Atl Nino/ENSO teleconnections. This paragraph now reads:

“More continuous and high-resolution proxy records from both the Pacific and Atlantic Ocean that are able to capture both SST mean state and variance are critically needed. These proxy archives will allow quantifying the relative contributions of forced changes in ENSO versus internal variability and better elucidate the climatic changes that occurred in the tropics throughout the Holocene.”

5) I would also recommend that the text focuses on the sensitivity experiments and bring more information on the role of vegetation and dust. Most of the mechanisms discussed in the text to explain the ENSO response to insolation forcing have already been extensively discussed in previous studies (I.E Brown et al, 2008, Zheng et al. 2008, Luan et al. 2012, An and Choi 2014, Karamperidou 2015, Roberts et al., 2015, and more recently Saint-Lu et al. 2016 for the energetics). The authors should refer to these manuscripts to tell how their mid-Holocene simulation compares to previous studies. In particular, the multi-model analysis of An and Choi 2014 provides an estimate of the feedback terms for mid-Holocene simulations. A rapid comparison with what has already been done would thus be welcome.

We feel that our main result did not properly come across. Previous studies have linked the changes in ENSO behavior in the Holocene either to the direct response to the solar orbital forcing in altering the tropical Pacific mean state or to enhanced trades winds associated changes in the Walker circulation due to stronger Asian monsoon. However, here we show that the changes in the spatio-temporal characteristics of the Atlantic Niño caused by the strengthening of the WAM are responsible for the shift in the Walker circulation.

Our results also suggest that the largest impact of the orbital forcing on ENSO variability is indirect and via its effect on the strength of the WAM.

Therefore, the additional role of vegetation and dust reduction is to further strengthen the WAM compared to the MH_{PMIP} experiment, which leads to stronger Atlantic Niño-like anomalies (and further decreases its variability). The dust reduction in our experiment does affect directly the eastern equatorial Pacific, increasing shortwave radiation at the surface (Fig. S1c, d). However, our set of sensitivity experiments lean more towards a dynamical explanation (further strengthen the WAM compared to MH_{GS}) than a direct radiative effect in explaining the difference between the MH_{GS+RD} and MH_{GS} experiments. The MH_{GS} simulation

already shows a large warming of the eastern Pacific where no changes in radiative forcing relative to MH_{PMIP} occur (Fig. S3), which is due to the westward shift of the Walker circulation caused by the WAM. We now discuss this aspect in the manuscript (see below).

Following the suggestion of the reviewer, we have now included a brief comparison with previous studies and add a discussion on the role of dust under vegetated Sahara. This part now reads:

“Most of previous modeling studies^{e.g., 21,37–43}, investigating the changes in ENSO variability in the MH did not provide a direct estimate of the changes in the BPF, but have shown ocean and atmospheric circulation changes very similar to those seen in our MH_{PMIP} simulation. The atmospheric circulation changes that lead to increased ENSO stability have been associated to the orbital-induced strengthening of the South Asian monsoon^{38,40}. Other studies^{e.g., 16} have pointed out that the reduction in ENSO variance in the MH is owing to the changes in the tropical Pacific mean state (asymmetric west-east temperature response) as a direct response to orbital forcing. However, our MH_{GS+RD} experiment shows that the dust reduction increases shortwave radiation at the surface the eastern equatorial Pacific (Fig. S1c, d) and hence possibly warming the SST there (Fig. S3). This should then decrease the east-west equatorial Pacific temperature gradient in the MH_{GS+RD} relative to the MH_{GS} . According to the mechanism proposed in Clement et al.¹⁶, the orbital forcing should favour a stronger response – larger heating – of the atmosphere on the western than on the eastern equatorial Pacific. Consequently, the reduced east-west temperature gradient due to the dust reduction-induced warming in the eastern Pacific in the MH_{GS+RD} should cause an increased rather than decreased ENSO variance relative to MH_{GS} . Our sensitivity experiments seem then to favor a dynamical explanation more than a direct radiative effect in explaining the difference between the MH_{GS+RD} and MH_{GS} experiments: the MH_{GS} simulation already shows a large changes in equatorial Pacific SST where no changes in radiative forcing relative to MH_{PMIP} occur (Fig. S3). A notable intensification and westward shift of the Walker circulation instead takes place in the MH_{GS} compared to MH_{PMIP} experiment (Fig. 4a, b). Our results indeed highlight that the anomalies in the Walker circulation, wind stress, thermocline, ocean current velocity and temperature stratification are proportional to the strength of the WAM (Fig. 5; $PI < MH_{PMIP} < MH_{GS} < MH_{GS+RD}$), suggesting that another mechanism than previously thought may be crucial in suppressing MH ENSO variability.”

We have now included also changes in the wind stress, thermocline depth, surface ocean current and temperature stratification for the MH_{GS} experiment in order to better understand the role played by the reduction of dust. We have now included this figure in the main text (Fig. 5; see also here below).

Finally, in order to further highlight the role of vegetation and dust vs. the orbital forcing alone, we have included the analysis and the discussion of the PI_{GS+RD} in the main manuscript. (see also response to comment #7).

The An and Choi paper did not explicitly quantify the ENSO feedbacks as we did in our figure 3. They only showed the mean state differences as have many other papers on the subject, which imply the possible coupling strength change. We

have therefore cited them together with other studies as shown above. The only modeling study that we are aware of that has explicitly performed the Bjerknes feedback calculation is Liu et al. 2014. We have compared our BPF in MH_{PMIP} with them and included the following sentences in the discussion of our results:

“The MH_{GS} and MH_{GS+RD} experiments show a larger decrease in BPF (17% for both) than in MH_{PMIP} simulation (11%) relative to PI (Fig. 3b), which is consistent with stronger divergent flow in the central eastern Pacific, larger thermocline anomalies and weaker upwelling displayed in summer (cf. Fig. 5b, f, l).

The reduction in the BPF in our MH_{PMIP} simulation is identical to that found in a previous modeling study performed with the Community Climate System Model, in which the BPF has been explicitly calculated (11% decrease)¹⁵”

Figure 5: Changes in equatorial Pacific Ocean monthly mean characteristics. (a, e, i) Time-longitude plot of the changes between PI and each MH experiment in climatological monthly eastward wind stress, **(b, f, l)** thermocline depth as captured by the 20°C isotherm, **(c, g, m)** surface ocean current velocity, and **(d, h, n)** temperature stratification (SST – T at 50 m depth), where T_{50m} is the temperature at 50 m depth. All

fields are averaged between 5°S and 5°N. The hatched areas indicate regions in which the changes are not significant at 95% confidence level assessed using a two-sided *t* test.

We have substantially re-organized the manuscript in order to include this additional material and to better convey the main message of our manuscript. The results section is restructured as follow:

- Reduced ENSO variability enhanced by Sahara greening

Here we present the results of ENSO variability changes and how the Sahara greening enhances them (-10% in MH_{pmip} → 25% MH_{gs+rd}).

- Westward shift and intensification of Walker circulation weaken ENSO variability

We discuss what is causing the ENSO variability changes (atmospheric circulation changes without yet linking to the Atlantic Niño) and we briefly compare our results with previous studies. We show that the Sahara greening and the dust reduction amplify the orbital forcing and we highlight the potential tight link between WAM strength and ENSO variability.

- Enhanced WAM triggers a “permanent” Atlantic Niño and alters the Walker circulation

Here, we focus our attention to the equatorial Atlantic in order to understand how the WAM strength is able to affect the ENSO variability. We discuss the changes in equatorial Atlantic mean state and variability. Hence, we link the WAM/Atlantic Niño to ENSO. We present the recent studies that suggest the link between Atlantic Niño and ENSO under present day conditions and then our new analysis that prove this link in our model.

After the Result section we provide also the

- Model-proxy comparison

We provide a brief model-proxy comparison also including the shortcoming in our model simulations and in the proxy records. In particular we extend the discussion to the mean state changes in both Pacific and Atlantic and linking them to our results.

6) Figure 4 as it stands brings nothing new compared to these previous studies. It should also show similar plots to highlight what is the additional effect of vegetation and dust compared to insolation alone.

Figure 4 (now Figure 8) was meant to schematically explain the novel proposed mechanisms. Although the mechanism was explained in the caption, we agree with the reviewer that the message did not come across properly. Therefore we have modified the figure in particular including key words to better guide the readers through the main mechanism. As mentioned above, the mechanisms at play in the MH_{PMIP}, MH_{GS}, and MH_{GS+RD} are identical, while the novel aspect is the link with the West African Monsoon strength and the associated development of a “permanent” Atlantic Niño.

Figure 8: Schematic representation of the mechanisms behind the changes in ENSO variability and mean state. (a) Changes in the strength of the WAM and location of the ITCZ in summer (JASO) (1) trigger an Atlantic Niño-like mean state and damp its variability (2). These conditions over the equatorial Atlantic shift the Walker circulation westward, forcing a divergent flow in the central Pacific (3). This flow cools the western Pacific and warms the eastern Pacific, with a consequent shallowing of the thermocline in the west and a deepening in the east (4) (vertical profile insert). The deepening of the thermocline and weakened upwelling in the eastern Pacific during the El Niño developing season reduces the strength of the Bjerknes feedback (4). **(b)** The thermocline anomalies in the central Pacific travel eastward reaching the eastern Pacific during winter favoring a climatological La Niña mean state in winter (5).

7) Also the feedback estimates should come with error bars, since some of the models have very large internal noise (cf Emile-Geay et al. 2016 for it) or diversity in ENSO characteristics (Wittenberg et al. 2009).

Thanks for the suggestion. We agree with the reviewer that ENSO has a very large internal noise and this was the reason why we had run long equilibrium simulations of 200 years for the sensitivity experiment and 575 years for the PI. We have now included the standard error of the mean for each simulation in the figure caption:

“The total BPF and the relative standard error of the mean are: $3.0 \pm 0.2 \text{ yr}^{-1}$ for the PI; $2.65 \pm 0.09 \text{ yr}^{-1}$ for the MH_{PMIP} ; and $2.5 \pm 0.2 \text{ yr}^{-1}$ for both MH_{GS} and MH_{GS+RD} experiments. The standard error of the mean has been calculated performing the BPF for sliding windows of 30 years and 10-year steps.”

7) Additional analyses could be provided on the mechanisms by which ENSO variability could be linked to WAP. The argument is based on the fact that both ENSO variability and tropical Atlantic variability are damped, using

results from a published sensitivity experiment on present day climate. Part of it could be better documented from their simulations.

In order to show the link between the Atlantic Niño and the ENSO we have performed a composite of Walker circulation changes associated to the Atlantic Niño phase in the PI simulation (Fig. 6). This analysis reinforces our previous results. We now present this result in the manuscript and it reads:

“Over western equatorial Atlantic the composite displays a remarkable strengthening of the convection in the PI simulation, which is consistent with increased precipitation over the Gulf of Guinea during positive Atlantic Niños (Fig. S7). In the MH simulations, such increase in convection over western equatorial Atlantic is much weaker and shallow (cf. Figs. 4 and 6) because of the prominent shift of the rain belt well into the Sahel/Sahara region and a consequent drying over the Gulf of Guinea (Fig. S2). The relative reduction in rainfall over the Gulf of Guinea associated with the northward expansion of the WAM and warming of eastern equatorial Atlantic SST is consistent with proxy evidence^{48,49} (Fig. 1a).”

Figure 6: Composite of Walker circulation anomalies associate to positive Atlantic Niño phases. PI climatological zonal stream function of the Walker circulation (contours: 0.2×10^{11} Kg/s interval from -1.4 to 1.4×10^{11} Kg/s; 0 line in bold) for the period June to October and its composite anomalies (positive minus negative) associated to Atlantic Niño phase. The zonal stream function composite has been calculated including the anomalies associated to Atlantic Niño and Niña events exceeding 1.5 standard deviations (0.69°C).

We have also tested the model to see whether the Pacific/Atlantic teleconnection was well represented compared to observation as also requested by the Reviewer (see answer to specific comment (a) below).

As mentioned above, we have also included in the main text the analysis and the discussion related to the PI_{GS+RD} showing that the development of the Atlantic Niño

and the changes in its spatio-temporal characteristics are able to affect the Walker circulation, the ENSO mean state and activity. It now reads:

“As a final test to further corroborate that the “permanent” Atlantic Niño is caused by the changes in atmospheric circulation (i.e., not a direct response to insolation) and is a major player for ENSO activity changes, we perform an additional simulation identical to the MH_{GS+RD} but with modern day insolation forcing (PI_{GS+RD}). In doing so, we isolate the effect of the Sahara greening and dust reduction. Our results show similar - albeit weaker - changes in the SST and precipitation pattern as those seen in the MH_{GS+RD} experiments (cf. Figs. 7a, b and S2f, S3f): a stronger WAM develops over Northern Africa and Atlantic Niño conditions are present (Figs. 7a, b). The Walker circulation is shifted westward (Figs. 7c) and La Niña-like conditions tend to develop in winter (Fig. S8). Finally, the PI_{GS+RD} simulation also shows a large reduction in the standard deviation of the Atlantic Niño index of ca. 40% and of the Niño 3.4 index of ca. 13%. This additional experiment helps to disentangle the effects of Saharan vegetation and dust reduction from the effect of insolation, highlighting the relative importance of the Sahara greening in affecting the ENSO activity. Our results suggest that the impact of orbital forcing on ENSO variability is to a large extent indirect and operates via its effect on the strength of the WAM”

Figure 7: Changes in surface climate and Walker circulation for the PI_{GS+RD} simulation. (a) Changes in SST (shadings) and SLP (contours: 0.25 hPa interval from -3.5 to 3.5 hPa; 0 value omitted for clarity) and **(b)** precipitation (shadings) and 10 m wind (vectors) for JASO in the PI_{GS+RD} simulation relative to PI. Only significant values at the 95% confidence level assessed using a two-sided t test are shaded. **(c)** PI climatological zonal stream function of the Walker circulation (contours: 0.2×10^{11} Kg/s interval from -1.4 to 1.4×10^{11} Kg/s; 0 line in bold) and associated changes (shadings) in the PI_{GS+RD} experiment relative to the PI for JASO.

8) The inclusion of the radiative effect of dust seems to partly counteract the effect of vegetation. Is there something to learn from the differences between these two simulations? What about the change in the west Pacific seen in the additional material figures? Is there a change in stratification related to the west Pacific wind (the additional material suggest that there isn't), or could those winds counteract the generation or propagation of Kelvin waves? There are differences between the simulation with vegetation and with dust that could also be used to point out the important factors.

As presented above the radiative effect of dust seems to play a small role. We have now included the detailed analysis of the changes in wind stress, thermocline, ocean currents and temperature stratification to address the reviewer comment (see answer to comment #5).

Minor points.

a) The text is very careful not overselling too much the results and recognizes that the sensitivity experiments represent extreme possible limit of W African vegetation and dust effect. A few words on the limits of the models in representing WAM/ENSO teleconnection should be mentioned, as well as previous finding of changes of this teleconnection in a different climate. Studies with previous model generation highlighted that ENSO/WAM teleconnection was too strong in most models (Jolly et al. 2007, Zhao et al. 2007). Could this affect the teleconnection mentioned here?

We have done a maximum covariance analysis (tropical SST and precipitation) in our model using observation/reanalysis data sets. We have included a discussion on the matter in the Method section. It now reads:

“Evaluation of the Atlantic/Pacific teleconnection in our model

Given the teleconnection between the tropical Atlantic and Pacific presented in this study, we deem relevant to test our model performances in capturing the ENSO/Atlantic Niño/WAM mode of variability. Previous studies^{66,67} have shown that although the models are able to reproduce the observed modes of variability, the impact of tropical SSTs on the WAM precipitation is generally overestimated. Here we use the maximum covariance analysis (MCA) to evaluate this teleconnection. The MCA can be considered as a generalization of the principal component analysis. The MCA analysis looks for patterns in two space-time datasets (SST and precipitation in our case), which explain a maximum fraction of the covariance between them. The MCA provides two sets of singular vectors and a set of singular values that are associated with each pair of vectors. Each pair of vectors represents a fraction of the squared covariance between the two variables (SCF: squared covariance fraction). The expansion coefficients for each variable are calculated by projecting the respective data field onto the respective singular vector. The correlation value (Corr.) between the expansion coefficients of the two variables indicates how strongly related the coupled patterns are.

We compared the MCA performed using the output of the PI simulation to MCA calculated based on the precipitation and SST data set provided respectively by the NOAA/OAR/ESRL PSD (Global Precipitation Climatology Project (GPCP) Combined Precipitation Data Set Version 2.2: <http://www.esrl.noaa.gov/psd/>) and the Hadley centre (HadISST⁶⁸) for the period 1979-2013.

The analysis shows that overall the model is able to reproduce the leading mode of covariance between the SST and precipitation pattern in North Africa: Atlantic Niño and La Niña conditions are associated with increased precipitation over the Gulf of Guinea. While our model shows a similar squared covariance fraction (62% vs. 67% in our model) for the leading mode of covariance as well as correlation between SST and precipitation over northern Africa (0.51 vs. 0.61), the precipitation anomaly is too strong, as in most climate models, and shifted southward compared to the reanalysis data sets (Fig. S7b and d). In our model the SST pattern shows weaker anomalies in the tropical Pacific and stronger in the Atlantic relative to the

reanalysis. The southward shift in our model is likely due to the WAM dry bias: the WAM's northernmost extent is located at 14.0° N about 250 km too far south compared to observation (see Fig. S6 and relative discussion in Pausata et al.²⁷). The strong correlation between the expansion coefficients associated to SST and precipitation time series has been interpreted as a modulation of the monsoon activity by tropical Pacific SST^{66,67}. However, the modulation of the monsoon activity in northern Africa is also caused by the Atlantic Niño phase, which in turn can affect the tropical Pacific SST.

In our model the SST pattern shows weaker anomalies in the tropical Pacific and stronger in the Atlantic relative to the reanalysis. This can be seen as a stronger influence of the tropical Pacific on the Atlantic SST or a more feeble impact of the Atlantic Niño on ENSO in our model compared to the reanalysis. Therefore, it may well be that the effects on ENSO activity during the MH associated to the “permanent” Atlantic Niño state may have been larger than those simulated. However, more in-depth analyses are needed to disentangle this aspect, which is beyond the scope of this study.”

Figure S7: (a-b) Maximum covariance analysis (MCA) of the sea surface temperature (SST) (left) and precipitation (right) and sea surface temperature (SST) (right) calculated using the observed/reanalysis GPCP Combined Precipitation Data Set Version 2.2 and HadISST; and (c-d) the model data. The squared covariance fraction (SCF) and the correlation values between the SST and precipitation expansion coefficients are provided at the top of the left panels.

b) The method or the additional material should provide addition information on the simulations and the way the model is set up for the different simulations. Are the authors sure that the model is exactly the same (I.E same feedback included) when they impose the vegetation and run with the dust model? How is dust considered in the different simulations, including the pre-industrial control?

We have now included additional information regarding the experiment designed. We agree with the reviewer that it is handier to have such information directly in

the paper rather than dig them out in Pausata et al. 2016. The dust as well as the vegetation cover and properties are prescribed therefore the model is identical, including the same feedbacks. We have now made this clear in the methods and it now reads:

“Boundary conditions for the mid-Holocene control (MH_{PMIP}), except for the orbital forcing and greenhouse gases, were set at pre-industrial values according to the PMIP3/CMIP5 protocol²². This includes land surface, aerosols, ice sheets, topography and coastlines. The orbital forcing was set at 6,000 years BP values and computed internally using the method of Berger⁶³. Differences in the Earth’s orbit in the MH enhanced the amplitude of the seasonal cycle in Northern Hemisphere insolation by ~5% compared to present day values. For the greenhouse gases we changed methane concentration from 760 ppmv PI value to 650 ppmv for MH according to PMIP/CMIP5 protocol, and kept CO₂ and other greenhouse gases the same as PI. Vegetation cover and properties, and dust concentrations are prescribed. The dust distribution used in this study and in Pausata et al.³⁰ was taken from the Community Atmosphere Model (CAM)⁶⁴, which is used in the Coupled Model Intercomparison Project (CMIP) phase 5. A second set of experiments is carried out in which the vegetation type over the Sahara domain (11°–33°N and 15°W–35°E) is set to shrub (MH_{GS}) and the PI dust amount is also reduced by up to 80% (Fig. 1 and S1 in Pausata et al.³⁰), based on recent estimates of Saharan dust flux reduction during the MH^{26,27} (MH_{GS+RD}). The vegetation change corresponds to a reduction in the surface albedo from 0.3 to 0.15 and an increase in the leaf area index from 0.2 to 2.6 (mainly desert and shrub respectively; Table 1 in ³⁰). The dust reduction leads to a decrease in the dust aerosol optical depth (AOD) of almost 60% and in the global total AOD of 0.02 (see Fig. 1 in Pausata et al.³⁰). The 80% dust reduction was applied over a broad area around the Sahara desert from the nearby Atlantic Ocean to the Middle East and throughout the troposphere (up to 150 hPa). A smoothing filter was used to avoid abrupt transitions in dust concentrations (see Fig. S1b, d, f in in Pausata et al.³⁰). Above 150 hPa the dust reduction was more evenly applied due to the fact that aerosol particles are uniformly distributed at those elevations. The change in dust concentration and vegetation cover are not meant to provide a faithful representation of the MH conditions over the Sahara and nearby regions, since no accurate vegetation reconstruction is available at the moment. They have instead been designed to more easily disentangle the effects of land surface cover and dust loading on atmospheric circulation.”

Response to Reviewer #3:

The only novel aspect is the links to the Sarah region, but this is still the results of just one model. The authors go into a bit of detail to try and explain how and why the changes impact ENSO, but all I keep thinking is would I find a similar result if I did the same experiment in a different model. As shown in the results of Bellenger et al. 2014 (in climate dynamics), while most models do a reasonable job of representing ENSO, there is a large amount of underlying error compensation occurring which raises the question are they getting it right for the right reasons.

The fact that the study has been done with just one model is not at all unusual, because it is not yet possible to perform model inter-comparison with the desired sensitivity experiments. Indeed, several pivotal modeling studies have been performed with only one model and published in high-profile journals, including the following: on the green Sahara: Kutzbach et al., *Nature*, 1996; Kutzbach and Liu, *Science*, 1997; Liu et al., *Nature*, 2014; on other topics from paleoclimate to effects of volcanic eruptions on climate, but still using only one model: Pausata et al., *Nature Geoscience*, 2011; Otterå et al., *Nature Geoscience*, 2010; Swingedouw et al., *Nature Communication*, 2015; Pausata et al., *PNAS*, 2015; Stevenson et al., *PNAS*, 2017.

Therefore, the point of these seminal studies is to stimulate other modeling groups in performing similar experiment to corroborate or not the new proposed mechanism.

In my opinion, the authors should aim for a longer format journal to better explain the details of the atmospheric teleconnections and how this leads to changes in ENSO.

Nature communications allows up to 5000 words and 10 display items in the main manuscript. In the revised version we have extended the discussion and provide further analyses to explain the atmospheric teleconnections as presented in the answers to the constructive suggestions of Reviewer #4.

REVIEWERS' COMMENTS:

Reviewer #4 (Remarks to the Author):

Compared to previous version the authors took into account most of my comments. The manuscript substantially improved. It is now clear that the discussion is on the sensitivity experiments and that the data are just there to check if the results make sense and not to compared to reality. Several figures and paragraphs have been added to better discuss the mechanisms. The addition of a simulation of the preindustrial climate with vegetated and less dusty Sahar-Sahel is interesting and convincing. However, I still have comments on the analyses and several points need clarification.

Major comments :

- The author should avoid the terms Niño, Niña, permanent Niño or Niña like when they discuss the mean state or the changes in seasonality. They are not the only one to do it, but it is misleading and confusing. I would therefore recommend to refer to upwelling,, seasonal cycle and annual mean were needed and not to these terms. This is particularly true for the section on permanent Atlantic Niño, in which it is difficult to isolate the part that refers to the mean state and the part that refers to the fact that interannual SST variability is reduced in the Gulf of Guinea.

- This above remark is important because the analogy between the change in the Walker circulation resulting from changes in the 6ka annual mean cycle (fig 4) or resulting from an El-Niño event is finally quite poor and not that convincing. The analogy holds for the Atlantic sector, the far west Pacific, but not the central and eastern Pacific. I am therefore not entirely convinced by the proposed arguments. There seems that the fact that the strengthening of the WAM induced by vegetation and dust strengthens the mean change in the Walker circulation and reinforces the mechanism induced by insolation alone is sufficient to make the point. I suspect that other factors than the SST warming in the Gulf of Guinea are important to explain the results. The story with the additional reduction of SST variability in the Atlantic, should be better quantify. How big is the magnitude of the changes in SST due to variability compared to the magnitude of the change in mean SST. If the mechanism involved for variability holds to explain the teleconnection for the mean state and the linkage with the development of El Niño event, then why is the reduced variability not partially counteracting the effect of the mean state and contributing to enhance ENSO variability?

- Several data sources claim for a "Niña like" mean state in the mid-Holocene. As in other simulations, the simulations presented here provide increased SST gradient only during DJF, whereas such east-west gradient doesn't show up in annual mean. The point is raised in the paper, but the conclusion states that the model produces a Niña like state in SJF like in the observations. This statement should be revisited.

Additional minor comments

1. 1st sentence. Remove "highly", remains uncertain is enough given the increase of number of records in the last years and the consistent picture they provide.

2. Make clear in the introduction that it is from highly idealized experiments. This doesn't change the objective and interest of the paper but indicate that the numbers provided should be considered as indicative and not as the truth.

3. End of introduction : slightly tone down the conclusion by stating "Saharan vegetation and dust emissions could have been critical factors" . This is important because even though the effect emerging from the simulations is very interesting and would receive lots of attention, the fact that it doesn't fit will with what happens at the end of the humid period suggests that it is only part of a more complex story.

4. l 72. Several models have interactive vegetation through interactive LAI and carbon cycle or dynamical vegetation in PMIP. The point is that they do not produce vegetation changes as far north as suggested by the observations.

- 5 l 73, do not forget the change in obliquity. Simulations also include changes in the atmospheric

trace gases, even though it is not a dominant factor here.

6 &l194_209: could also mention that An and Choi 2014 invoke the fact that water vapor feedback is reduced because of the colder tropics (in annual mean)

7. L2019 I am not sure proportional is the right term.

8. L276. Revisit the description and provide some indication of the locations where it matches

9. L 342-342 : LGM is misleading here. The first effects are certainly due to the ice-sheet and colder tropics. These sentences should be removed.

10. figure 5: May be adding isolines for the PI climate as in figure 4 would be welcome.

Response to Reviewer #4:

Major comments:

- The author should avoid the terms Niño, Niña, permanent Niño or Niña like when they discuss the mean state or the changes in seasonality. They are not the only one to do it, but it is misleading and confusing. I would therefore recommend to refer to upwelling,, seasonal cycle and annual mean were needed and not to these terms. This is particularly true for the section on permanent Atlantic Niño, in which it is difficult to isolate the part that refers to the mean state and the part that refers to the fact that interannual SST variability is reduced in the Gulf of Guinea.

We have edited the section related to the permanent Atlantic Nino removing all reference to permanent Atlantic Nino and Nino/Nina-like conditions.

- This above remark is important because the analogy between the change in the Walker circulation resulting from changes in the 6ka annual mean cycle (fig 4) or resulting from an El-Niño event is finally quite poor and not that convincing. The analogy holds for the Atlantic sector, the far west Pacific, but not the central and eastern Pacific. I am therefore not entirely convinced by the proposed arguments. There seems that the fact that the strengthening of the WAM induced by vegetation and dust strengthens the mean change in the Walker circulation and reinforces the mechanism induced by insolation alone is sufficient to make the point. I suspect that other factors than the SST warming in the Gulf of Guinea are important to explain the results. The story with the additional reduction of SST variability in the Atlantic, should be better quantify. How big is the magnitude of the changes in SST due to variability compared to the magnitude of the change in mean SST. If the mechanism involved for variability holds to explain the teleconnection for the mean state and the linkage with the development of El Niño event, then why is the reduced variability not partially counteracting the effect of the mean state and contributing to enhance ENSO variability?

While we agree with the reviewer that “the fact that the strengthening of the WAM induced by vegetation and dust strengthens the mean change in the Walker circulation and reinforces the mechanism induced by insolation alone is sufficient to make the point”, we are trying to provide a mechanism through which the WAM is able to affect the ENSO. Although we can't fully prove it, we provide a set of analyses and previous studies that support our mechanism. Further studies are with no-doubt needed to further investigate it.

In addition, the similarity in the Walker circulation shift between the ATL Nino

cases and the MH experiments is also present in the central Pacific. We agree with the reviewer that besides the warming in the Gulf of Guinea there is also the variability change while figure 5 only shows the SST warming. Furthermore, figure 1c in Li et al. 2016 which is based on observation also show a similar change in the wind patterns compared to our MH simulation.

We have included a quantification of the SST mean state change through the strength of the east-west gradient in the equatorial Atlantic and we have included it in Table 1 and in the text:

“Our model shows an increase SST seasonal cycle with a warming taking place from late summer to late winter/early spring. The strength of the warming seems to be related to the intensity of the WAM (Supplementary Figure 2). The JASO east-west temperature gradient drops of 97%, 250% and 225% in the MH_{PMIP} , MH_{GS} and MH_{GS+RD} respectively (Table 1).”

Regarding the comment on the mechanism involved for variability we are not sure we have understood the reviewer point. In the first part we agree that if the Atlantic affects ENSO through atmospheric teleconnection this should be valid for any process longer than a months (atmospheric adjustment time scale). Therefore, reduced ATL variability reduces ENSO through atmospheric teleconnection. However, we don't understand why this should counteract the mean state effect and enhance ENSO.

Perhaps, we could make the point a clearer: the variability acts as a source of external noise forcing that directly forces variability in the ENSO region, while the mean state changes the stability.

Mathematically, in terms of delayed oscillator: $dT/dt = aT(t) - bT(t-\tau) + \text{Noise}$
Atlantic variability reduces the noise forcing and in turn reduces ENSO mean state reduces coupled instability a , and reduces ENSO.

Both will not counteract, but enhances each other. We have included the following paragraph also in light of the comment related to the quantification of mean state vs. variability impacts on ENSO activity.

“Our results show that changes in the variability equatorial Atlantic SST are sources of external noise forcing that directly influence variability in the ENSO region, while changes in the mean state (Atlantic Niño anomaly) affect ENSO stability through changes in the Walker circulation. Therefore, they act in synergy in perturbing ENSO activity. Our experiments show that ENSO variability does reach its smallest values in the MH_{GS+RD} when Atlantic Niño variability is minimal and the equatorial Atlantic temperature gradient is strongly reduced relative to the PI. However, ENSO activity in the MH_{GS+RD} is only 2% lower than in the MH_{GS} experiment – albeit a notable difference in Atlantic Niño variability (9% difference, Table 1). This is likely because the equatorial Atlantic temperature gradient is instead less reduced in the MH_{GS+RD} than in the MH_{GS} experiment (25%), therefore partially counteracting the decrease in Atlantic Niño variability (Table 1). The increase in the MH_{GS+RD} temperature gradient compared to MH_{GS} is

likely due to reduced dust emissions that tends to warm the western side of the equatorial Atlantic (Supplementary Figure 3e, f)."

- Several data sources claim for a "Niña like" mean state in the mid-Holocene. As in other simulations, the simulations presented here provide increased SST gradient only during DJF, whereas such east-west gradient doesn't show up in annual mean. The point is raised in the paper, but the conclusion states that the model produces a Niña like state in SJF like in the observations. This statement should be revisited.

We have accordingly modified the paragraph, providing a more accurate description and comparison with proxy archive in which we highlight the discrepancy and also point out that the proxy records may have also shift the preferred growing season relative to today:

"In regard to changes in SST and precipitation mean state in the tropical Pacific, proxy archives seem to confirm a westward shift in the Walker circulation with increased precipitation and warmer SST in the western Pacific warm pool during the MH relative to modern climate (Figure 1). The MH proxy data also indicate an enhanced zonal SST gradient across the equatorial Pacific, with SST cooling in the eastern part of the basin and warming in the west (Figure 1). In contrast, our model simulates an annual net warming on both sides of the basin (Supplementary Figure 10). Nevertheless, our modeled MH seasonal SST gradients do reveal a close match between the proxy records and the modeled SST gradient during boreal winter/early spring. It is possible that some proxy-based reconstructions of equatorial Pacific SSTs could reflect specific seasons rather than annual means, as suggested by previous studies⁵¹⁻⁵³. For example, cooling of the warmer months and warming of the colder months in the equatorial Pacific during the MH may have shifted the preferred growing season relative to today."

Additional minor comments

1. 1st sentence. Remove "highly", remains uncertain is enough given the increase of number of records in the last years and the consistent picture they provide.

Done.

2. Make clear in the introduction that it is from highly idealized experiments. This doesn't change the objective and interest of the paper but indicate that the numbers provided should be considered as indicative and not as the truth.

We have highlighted in the introduction that the simulations are idealized. We prefer to use "idealized" rather than "highly idealized" because as discussed in Pausata et al. 2016 and as suggested by a recent paper of Tierney et al. 2017 shrub vegetation was likely over most part of western Sahara (at least). The amount of dust reduction is also based on recent estimate of dust changes during the African Humid Period (McGee et al. 2013).

"We analyze a set of idealized climate model simulations in which prescribed

Saharan vegetation and dust concentrations are changed in order to investigate the hitherto unexplored impacts of Sahara greening on the spatio-temporal characteristics of ENSO during the MH."

In the model-proxy comparison we stress it again:

"Even if such data were available, the highly idealized nature of our simulations may not capture the complex evolution of tropical Pacific variability during the Holocene."

3. End of introduction : slightly tone down the conclusion by stating "Saharan vegetation and dust emissions could have been critical factors". This is important because even though the effect emerging from the simulations is very interesting and would receive lots of attention, the fact that it doesn't fit well with what happens at the end of the humid period suggests that it is only part of a more complex story.

We have rephrased it as follow:

"Our results thus suggest that Saharan vegetation and dust emissions are critical factors in amplifying ENSO's response to insolation forcing, suggesting that potential changes in the WAM due to anthropogenic warming may influence ENSO variability in the future, as well."

4. I 72. Several models have interactive vegetation through interactive LAI and carbon cycle or dynamical vegetation in PMIP. The point is that they do not produce vegetation changes as far north as suggested by the observations.

The majority of PMIP2 and PMIP3 models have prescribed vegetation and few have interactive vegetation. However, models using interactive vegetation start the experiment from pre-industrial boundary conditions and 6ka insolation, which is already weak compared to 10 ka when the maximum of insolation occurred. As discussed in Pausata et al. 2016:

"Studies using climate models coupled to a dynamical vegetation also present a broad range of results. For example, Claussen and Gayler (1997) and Renssen et al. (2006) found very strong precipitation increases due to the greening of the Sahara – respectively ~50% and ~25% greater than in our study (MHGS-PD). Claussen and Gayler (1997) simulated a mean June-to-August precipitation of 129 mm/month (over 10°W–30°E and 15°N–30°N); while in our simulation the mean precipitation in the same domain is ~86 mm/month. Furthermore, their MH reference experiment was twice as wet as our MHCNTL simulation. Renssen et al. (2006) showed an increase in rainfall over the Western Sahara (14°W–3°E) of approximately 45–50 mm/month around 26–27°N, whereas in our simulation (MH_{CS}) the increase is around 38 mm/month. In Levis et al. (2004), the rainfall increase is remarkably smaller in the equatorial region (up to 15°N) compared to our study (MH_{CS}), nevertheless the WAM penetrates further north than in the MHGS-PD experiments when considering vegetation and soil feedbacks (cf. Fig. 3 with Fig. 1 in Levis et al., 2004). At the opposite end of the spectrum, other works

show weaker precipitation enhancements (e.g., Braconnot et al., 1999; De Noblet-Ducoudre et al., 2000) compared to the present study. In particular, the most recent simulations performed with HadGEM2-ES and MIROC-ESM, which included both a dynamic vegetation and dust emissions, were not able to reproduce a comparable strengthening of the African monsoon (Harrison et al., 2015). Those simulations were submitted to the CMIP5 initiative and the initial configuration for the vegetation was therefore set to pre-industrial levels. The simulated MH changes in both models do not allow vegetation to grow above 15°N (see simulated precipitation anomalies in Fig. 2 in Harrison et al., 2015). Hence, the associated variations in dust load compared to pre-industrial climate are most likely limited, and so are the feedbacks.”

5 | 73, do not forget the change in obliquity. Simulations also include changes in the atmospheric trace gases, even though it is not a dominant factor here.

Thanks, we have changed it in order not to repeat twice the precessional forcing (which correspond to changes in the seasonality of insolation) and point out the obliquity. It now reads:

“to changes in obliquity forcing alone and the seasonality of insolation.”

6 &194_209: could also mention that An and Choi 2014 invoke the fact that water vapor feedback is reduced because of the colder tropics (in annual mean)

We have included it in that paragraph:

“Other studies have pointed out that the reduction in ENSO variance in the MH is owing to the changes in the tropical Pacific mean state (asymmetric west-east temperature response¹⁶ or reduced water vapor feedback⁴²) as a direct response to orbital forcing.”

7. L219 I am not sure proportional is the right term.

We have added “somehow proportional”:

“Our results indeed highlight that the anomalies in the Walker circulation, wind stress, thermocline, ocean current velocity and temperature stratification are somehow proportional to the strength of the WAM.”

8. L276. Revisit the description and provide some indication of the locations where it matches

We have changed it to read:

“The analysis shows a net westward shift of the Walker circulation associated with positive phases of the Atlantic Niño that overall resembles the shift seen in MH simulations, in particular over western and central Pacific and western Atlantic.”

9. L 342-342 : LGM is misleading here. The first effects are certainly due to the ice-sheet and colder tropics. These sentences should be removed.

We have removed it.

10. figure 5: May be adding isolines for the PI climate as in figure 4 would be welcome.

Given the presence of the crosses we feel that the isolines would overwhelm the figures. For this reason we have, instead, provided a colorscale with relatively few colors in order to clearly discern them.